# Determining the bacterial cell biology of Planctomycetes

Christian Boedeker[1], Margarete Schüler[2], Greta Reintjes[3], Olga Jeske[1], Muriel C.F. van Teeseling[4,5], Mareike Jogler[1], Patrick Rast[1], Daniela Borchert[1], Damien P. Devos[6], Martin Kucklick[7,8], Miroslava Schaffer[2], Roberto Kolter[9], Laura van Niftrik[4], Susanne Engelmann[7,8], Rudolf Amann[3], Manfred Rohde[7], Harald Engelhardt[2] & Christian Jogler[1,4]

Bacteria of the phylum Planctomycetes have been previously reported to possess several features that are typical of eukaryotes, such as cytosolic compartmentalization and endocytosis-like macromolecule uptake. However, recent evidence points towards a Gram-negative cell plan for Planctomycetes, although in-depth experimental analysis has been hampered by insufficient genetic tools. Here we develop methods for expression of fluorescent proteins and for gene deletion in a model planctomycete, Planctopirus limnophila, to analyse its cell organization in detail. Super-resolution light microscopy of mutants, cryo-electron tomography, bioinformatic predictions and proteomic analyses support an altered Gram-negative cell plan for Planctomycetes, including a defined outer membrane, a periplasmic space that can be greatly enlarged and convoluted, and an energized cytoplasmic membrane. These conclusions are further supported by experiments performed with two other Planctomycetes, Gemmata obscuriglobus and Rhodopirellula baltica. We also provide experimental evidence that is inconsistent with endocytosis-like macromolecule uptake; instead, extracellular macromolecules can be taken up and accumulate in the periplasmic space through unclear mechanisms.

[1] Leibniz Institute DSMZ, Inhoffenstraße 7b, 38124 Braunschweig, Germany. [2] Department of Molecular Structural Biology, Max Planck Institute of Biochemistry, Am Klopferspitz 18, 82152 Martinsried, Germany. [3] Department of Molecular Ecology, Max Planck Institute for Marine Microbiology, Celsiusstraße 1, 28359 Bremen, Germany. [4] Department of Microbiology, Radboud University, Heyendaalseweg 135, NL-6525 AJ Nijmegen, Netherlands. [5] Department of Cellular Microbiology, Philipps-University Marburg, Faculty of Biology, Hans-Meerwein-Straße 4, 35043 Marburg, Germany. [6] Department of Cell biology and Biotechnology, CABD, Pablo de Olavide University-CSIC, Carretera de Utrera km1, 41013 Sevilla, Spain. [7] Helmholtz Center for Infection Research GmbH, Inhoffenstraße 7, 38124 Braunschweig, Germany. [8] Department of Microbial Proteomics, Technical University Braunschweig, Institute for Microbiology, Spielmannstraße 7, 38106 Braunschweig, Germany. [9] Department of Microbiology and Immunobiology, Harvard Medical School, Boston, Massachusetts, 02115, USA. Correspondence and requests for materials should be addressed to C.J. (email: Christian@Jogler.de).

For decades, Planctomycetes appeared to blur the line between prokaryotes and eukaryotes[1]. These unusual bacteria are ubiquitous and both environmentally[1] and biotechnologically[2,3] important. Planctomycetes were initially described as eukaryotes[4] and later acknowledged as bacteria[5], and they were proposed to share some sort of evolutionary link with eukaryotes[1,6,7]. This hypothesis was supported by conspicuous traits of the planctomycetal cell biology, traits which are usually associated with eukaryotes rather than with bacteria[1].

For example, Planctomycetes were thought to have a proteinaceous cell wall[8,9], while a peptidoglycan cell wall to maintain cell integrity is the bacterial hallmark trait. In addition, Planctomycetes are thought to comprise a compartmentalized cytosol that is separated *via* an intracytoplasmic membrane into a paryphoplasm and a pirellulosome[10]. However, the cytosol of bacteria is, with few exceptions[11], an enclosed compartment that follows the outer shape of the cell; the cytosol of eukaryotes is divided into multiple compartments (Fig. 1). Furthermore, some species of Planctomycetes (the so-called anammox bacteria) contain an anammoxosome, a distinct compartment for energy production that has been called a 'bacterial mitochondrion'[11,12]. Other planctomycetal species such as *Gemmata obscuriglobus* were reported to enclose their highly condensed nucleoid in double membranes, which structurally resembled the eukaryotic nuclear membrane[13]. Accordingly, a spatial separation of transcription and translation was suggested[14].

Uniquely, Planctomycetes and some other members of the Planctomycetes-Verrucomicrobia-Chlamydiae superphylum are the only organisms among bacteria and archaea whose genomes encode proteins with high-structural similarity to eukaryotic membrane coat proteins (MCs)[15]. Paralleling their eukaryotic counterparts, planctomycetal MCs comprise β-propeller domains followed by stacked pairs of α-helices[15]. A significant number of planctomycetal MC-like proteins are localized in close proximity to either the intracytoplasmic membrane (Fig. 1) or to vesicles within the paryphoplasm[15,16]. Eukaryotic MCs such as clathrins play a major role in the formation of coated vesicles during endocytosis and display similar localizations to those in Planctomycetes[14]. Endocytosis is considered as a hallmark of eukaryotes because it is presumed to have paved the way for the acquisition of an endosymbiont. Interestingly, planctomycetal vesicles have been suggested to support endocytosis-like uptake of macromolecules into the paryphoplasm of the planctomycetal species *Gemmata obscuriglobus*[15]. This would be the only vesicle-based uptake system to be found outside of the eukaryotic domain[16,17]. This served as a strong argument for an evolutionary link between Planctomycetes and eukaryotes.

Phylogenetically, the Planctomycetes are Gram-negative bacteria[18]. Nevertheless, the previously proposed planctomycetal cell plan differs significantly from that of other Gram-negative bacteria[1] and has been frequently revisited[19]. For example, the *G. obscuriglobus* 'nucleus' has been questioned[20]. Peptidoglycan—the hallmark of free-living bacteria—has been found in several Planctomycetes[21,22]. Recent bioinformatic and chemical analyses support a more typical Gram-negative cell plan than previously thought[23,24]. However, to the best of our knowledge, endocytosis-like uptake has not been experimentally tested again after the initial report[16]. Furthermore, research on planctomycetal cell biology is hampered by a paucity of genetic tools[25,26].

Here, we revisit planctomycetal cell biology using both existing[25,26] and new genetic tools, together with super-resolution light microscopy, bioinformatic predictions and proteomic analysis. Furthermore, we analyse frozen-hydrated planctomycetal cells with cryo-electron tomography and find further evidence for a Gram-negative cell plan that differs from that of other bacteria by the presence of an enlarged periplasmic space. Finally, we study the planctomycetal endocytosis-like process in detail.

## Results

**Light microscopy of the planctomycetal membrane organization.** To analyse the planctomycetal cell plan with fluorescence microscopy, we constructed a constitutive *gfp*-expressing *Planctopirus limnophila* strain (formally known as *Planctomyces limnophilus*), which was then stained with DAPI (DNA) and FM4–64 (lipid membrane). The DNA of *P. limnophila* was always condensed (Fig. 2a–j, DAPI: blue) while FM4–64 staining varied between individual cells. After analysing 1,838 bacteria, two different membrane staining patterns could be distinguished (representative overview shown in Fig. 2a–j; detailed analysis in Supplementary Fig. 1a,b); type 1 cells (27.6%) displayed a membrane staining pattern comparable to *E. coli* which served as 'typical' Gram-negative control (Fig. 2b,e,h and Supplementary Fig. 1c). Only the outer rim of these cells was stained red, indicating close proximity of the outer- and cytoplasmic membranes (Fig. 2e). In contrast, type 2 cells (72.4%) comprised additional red foci either within the cell or attached to the outer membrane (Fig. 2c,d,f,g). The GFP protein localized within the cytosol (Fig. 2c,d,f,g, GFP: green); cells were framed by a red stained membrane, while their interior was mostly green with some yellow areas (Fig. 2i,j). The yellow regions resulted from green cytosolic GFP and red membrane signals within the same area of the cell which becomes evident if intensity histograms of the three representative overlays are plotted. The additional maxima of such curves likely originate from invaginations of the innermost membrane into the cytosol (Fig. 2f,g, green and red arrowheads). If such membrane invaginations were smaller in diameter than the diffraction limit of light (about 200–250 nm), they would be visible as red dots in wide field (WF) microscopy. Furthermore, since FM4–64 preferentially stains the cytoplasmic membrane[27], this result implies that the innermost membrane is consistent with the cytoplasmic membrane. To prove this view, we treated both *P. limnophila* and *E. coli* cells with high sucrose concentrations, so that the ensuing plasmolytic effect could be observed in phase-contrast[28]. We could reproduce these findings in a *gfp*-expressing *E. coli* mutant (Supplementary Fig. 1c,d). After the treatment with 30% sucrose, additional staining revealed a significantly shrunken nucleoid (Supplementary Fig. 1d,i, *t*-test: $P = 0.0001$) and cytoplasm (Supplementary Fig. 1d,j, *t*-test: $P = 0.0001$) while the periplasmic space was enlarged (Supplementary Fig. 1d: phaco). When the same treatment was applied to *P. limnophila*, the proposed invaginations of the innermost membrane into the cytosol increased significantly (Supplementary Fig. 1l, *t*-test: $P = 0.0001$) and they became visible as distinct indentations (Supplementary Fig. 1f,h). Remarkably, the condensed nucleoid was even further compressed (Supplementary Fig. 1k, *t*-test: $P = 0.0009$) and an empty area became visible within the green GFP signal that illustrates an extreme enlargement of the periplasmic space (Supplementary Fig. 1f,h). This observation became even more evident when intensity histograms of untreated and treated *P. limnophila* cells were compared (Supplementary Fig. 1g,h). While untreated cells comprise a maximum of the GFP signal at mid cell (Supplementary Fig. 1g), treated cells show a local minimum at the same position (Supplementary Fig. 1h). To further verify our model (Fig. 2h–j: Model), we performed structured illumination microscopy (SR-SIM). This technique roughly doubles the resolution of WF microscopy (Supplementary Fig. 2a–c) and accordingly resolved the invaginations of the innermost membrane (Fig. 2k–m).

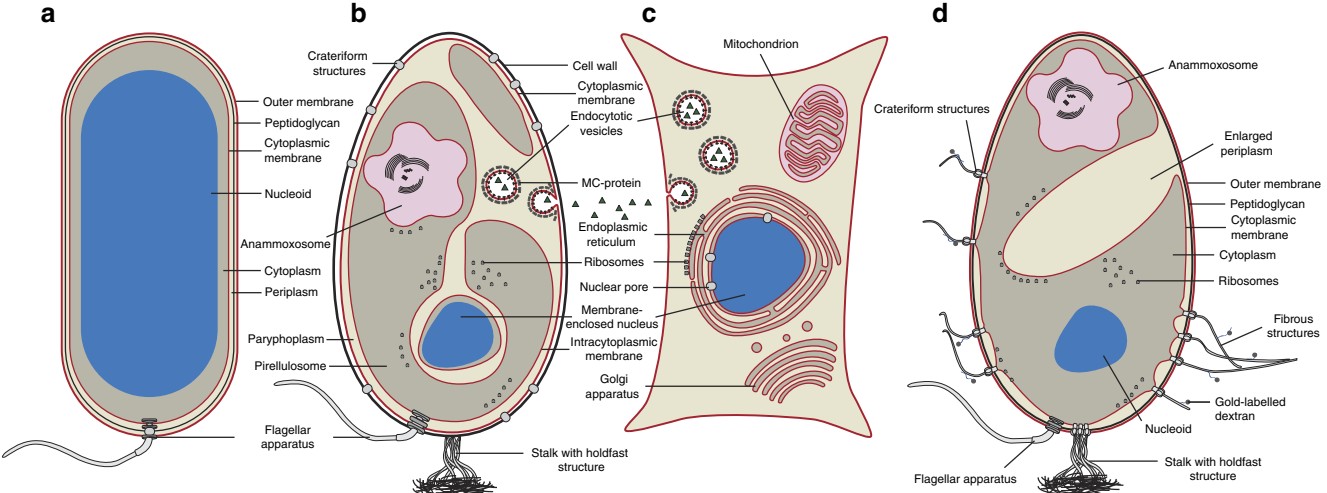

**Figure 1 | Cellular structures of a typical bacterium, planctomycetes and eukaryotes.** (**a**) A typical Gram-negative prokaryote is surrounded by an outer membrane (OM), a peptidoglycan (PG) cell wall and the cytoplasmic membrane (CM). The DNA forms the nucleoid and occupies a major portion of the cytoplasm. (**b**) Planctomycetes have been previously proposed to possess a unique cell plan. It was thought that PG was absent and replaced by a proteinaceous cell wall instead. The outermost membrane (OM) has been interpreted as CM, while an additional intracytoplasmic membrane (ICM) would divide the cytoplasm into a paryphoplasm and a pirellulosome. While the nucleoid of all Planctomycetes is highly condensed, *G. obscuriglobus* was proposed to contain an additional double membrane surrounding the DNA, similarly to the eukaryotic nucleus. Other planctomycetal species, the anammox bacteria, have additional subcellular structures such as the anammoxosome, an organelle responsible for the generation of energy. Most strikingly, Planctomycetes were reported to perform endocytosis-like uptake of macromolecules employing membrane-coat-like proteins that structurally resemble eukaryotic membrane-coat proteins such as clathrin. (**c**) A typical eukaryotic cell with membranous organelles and the ability to perform endocytosis. (**d**) Recent work and this study substantiate the view that Planctomycetes possess a Gram-negative cell architecture. The cells show a remarkable tendency for massive invaginations of the cytoplasmic membrane. Crateriform structures are found at sites of contact between the inner and outer membrane. Uptake of large molecules does not appear to be mediated by vesicles in *P. limnophila* and *G. obscuriglobus*.

Two other model Planctomycetes, *Gemmata obscuriglobus* and *Rhodopirellula baltica*, were analysed with similar staining experiments (Supplementary Fig. 2d–g). These species were chosen since they cover the planctomycetal phylogeny in a representative manner[20]. Given the larger size of *G. obscuriglobus*, the phase contrast images revealed more detail compared to *P. limnophila* and SR-SIM resolved the invaginations of the innermost membrane more clearly (Supplementary Fig. 2d,e). Similar results were obtained for *R. baltica* (Supplementary Fig. 2f,g). Taken together, our results, showing that FM4–64 and Nile Red stain preferentially the innermost membrane of Planctomycetes, supports the conclusion that this membrane represents the cytoplasmic membrane. This finding was further supported by SEM analysis (Supplementary Fig. 2h,i).

**Cryo-electron tomography of *P. limnophila*.** *P. limnophila* cells investigated by cryo-electron tomography displayed the two types of membrane invaginations, i.e., cells with a Gram-negative typical shape of the cytoplasmic compartment (Supplementary Fig. 3a–f) and cells with distinct invaginations, with a complex membrane organization (Fig. 3 and Supplementary Fig. 3g–l). Selected slices of the tomograms showed cytoplasmic invaginations, but we did not find any distinct 50–200 nm endocytic vesicles. Instead, these invaginations were still connected below and/or above in z-direction. Sometimes connections were formed by small cytoplasmic 'bridges' (about 50 nm in diameter), or by thin tube- or disk-like invaginations of the cytoplasmic membrane (Fig. 3, Supplementary Fig. 3 and Supplementary Movie 1). The latter could even convey the impression of a double membrane structure (Fig. 3). Yet, we did not observe distinctly isolated membranous compartments in our tomograms in addition to the common compartments cytoplasm and periplasm.

We would expect to find such endocytic vesicles if they are regular or frequent cellular structures, particularly as the cells were randomly oriented and we should not have systematically missed characteristic cellular zones. Moreover, we found only two types of density patterns in tomograms, either originating from the cytoplasm, including various structures of high contrast, or from the periplasm. Given that most of the reconstructed cells were thinned by focused ion beam (FIB) micromachining (remaining thickness 250–600 nm), we cannot absolutely exclude the possibility of other rare diversifications of the cellular membrane system.

The 3-D data illustrate how the complicated membrane shape is stabilised (Fig. 3m,n). The crateriform structures in the outer membrane, or at least some of them, are connected to the cell membrane and appear to fasten it in a position close to the cell wall. Where crateriform complexes are missing, the membrane can invaginate. Sometimes only a small membrane patch and a tubular volume of cytoplasm remained attached. The density of crateriform structures was $29.1 \pm 5.4$ per $\mu m^2$ for the total cell surface, and $40.2 \pm 12.8$ per $\mu m^2$ for the surface area where the cell membrane was not invaginated (data from 5 tomograms with 141 crateriform structures in total). The extrapolated number of protein complexes per cell was $104 \pm 17$.

**Membrane potential staining and ATPase localization.** If the innermost membrane is the genuine cytoplasmic membrane (as indicated by the FM4–64 staining experiments), it should be energized and contain the proton-translocating ATPases. To test this hypothesis, we stained all three model Planctomycetes with $DiOC_6(3)$ to visualize the membrane potential[29]; FM4–64 lipid staining was performed in addition to DAPI (DNA) staining. At least 50 cells per species were observed, and

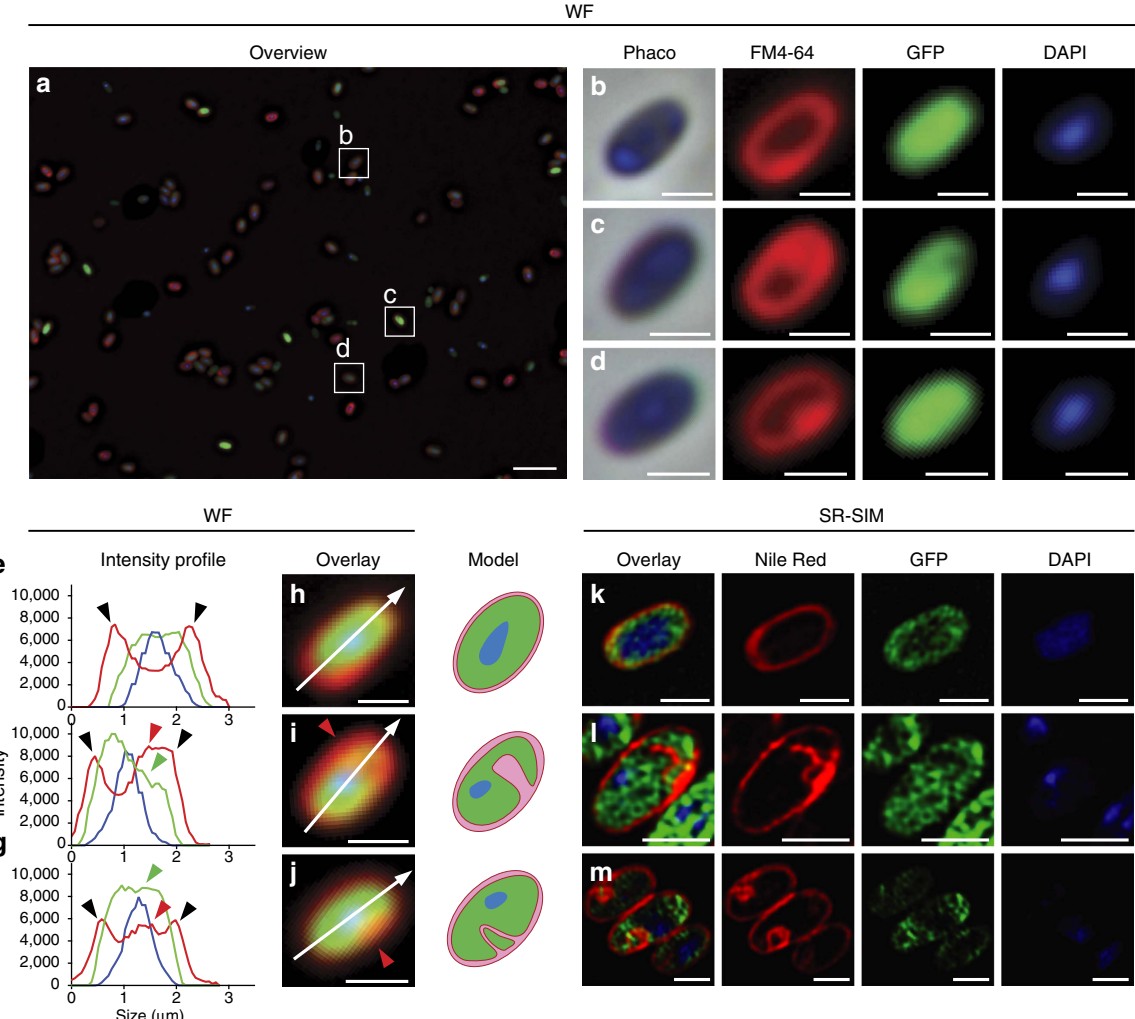

**Figure 2 | *Planctopirus limnophila* cells possess diverse enlargements of the periplasmic space.** (**a**–**m**) *P. limnophila* cells were engineered to produce GFP as cytoplasmic marker (green), while nucleoids and membranes were stained blue (DAPI) and red (wide field (WF): FM4–64, super resolution structured illumination microscopy (SR-SIM): Nile Red), respectively. 1,838 cells in 10 fields of view were analysed employing wide field microscopy (**a**: overview of one field of view; see Supplementary Fig. 1a,b for details). Two distinct phenotypes were observed: 27.6% of the cells comprised a typical Gram-negative membrane and cytoplasm staining patterns (**b**); 72.4% of the analysed cells showed additional red foci (**c,d**). Line profiles of fluorescence intensity were plotted longitudinal (**e**–**j**: along white arrows) and either two (**e**: black arrowheads) or multiple (**f,g**: red arrowheads) membrane staining maxima (**e,f**: red lines) were observed. The GFP signal maxima (**e**–**g**: green lines) never overlapped with the red outer membrane maxima (black arrowheads) while it did with additional membrane maxima (**f,g**: green and red arrowheads). If representative cells are considered (**b**–**d**), the corresponding overlays in (**i,j**) show colocalization (yellow) of the cytoplasm (green) and the membrane (red) signals in such cells. SR-SIM microscopy resolves these foci as invaginations of the innermost membrane into the cytoplasm (**k**–**m**: overlay and Nile Red) as schematically illustrated in the model. Scale bars, 5 μm (**a**) and 1 μm (**b**–**m**).

representative micrographs are shown in Supplementary Fig. 4a–f. The green $DiOC_6(3)$ stain is visible within the cells and co-localizes with red signals of the membrane stain. However, the red membrane stain always surrounds the outer rim of the cells where no green signal is observed (Supplementary Fig. 4). Colocalization was further verified employing intensity blots and calculating Pearson's correlations and Mander's overlaps (Supplementary Fig. 4g–i). These results denote that only the innermost membrane is energized.

In addition, we localized the ATPase within cells using WF fluorescence and direct stochastical optical reconstruction microscopy (dSTORM, Fig. 4a–d). ATPases were identified using an anti-$Na^+$-$F_1F_0$-ATPase rabbit antibody. The secondary anti-rabbit IgG (labelled with Alexa Fluor 647) led to rather diffuse red foci in WF micrographs (Fig. 4a). Given the size of both antibodies and the WF diffraction limit, fluorescence signals within 400 nm of a cell's outer rim cannot distinguish between ATPase localization at the innermost and outermost membrane. However, 60.79% of the detected signals came from the middle of the cell and are thus most likely related with ATPases that sit in the innermost (cytoplasmic) membrane which invaginated into the cytoplasm. 39.21% of the fluorescence signals resulted from the outer rim of cells and thus originates from either an ATPase localization at the outer membrane, or from a localization at the innermost cytoplasmic membrane (Fig. 4a,b, Supplementary Fig. 5a,b). dSTORM experiments revealed that the fluorescence signal was localized within the cell, and not exclusively around the outer rim as one would expect if the outermost membrane were energized (Fig. 4c,d). We further verified the localization of ATPases employing

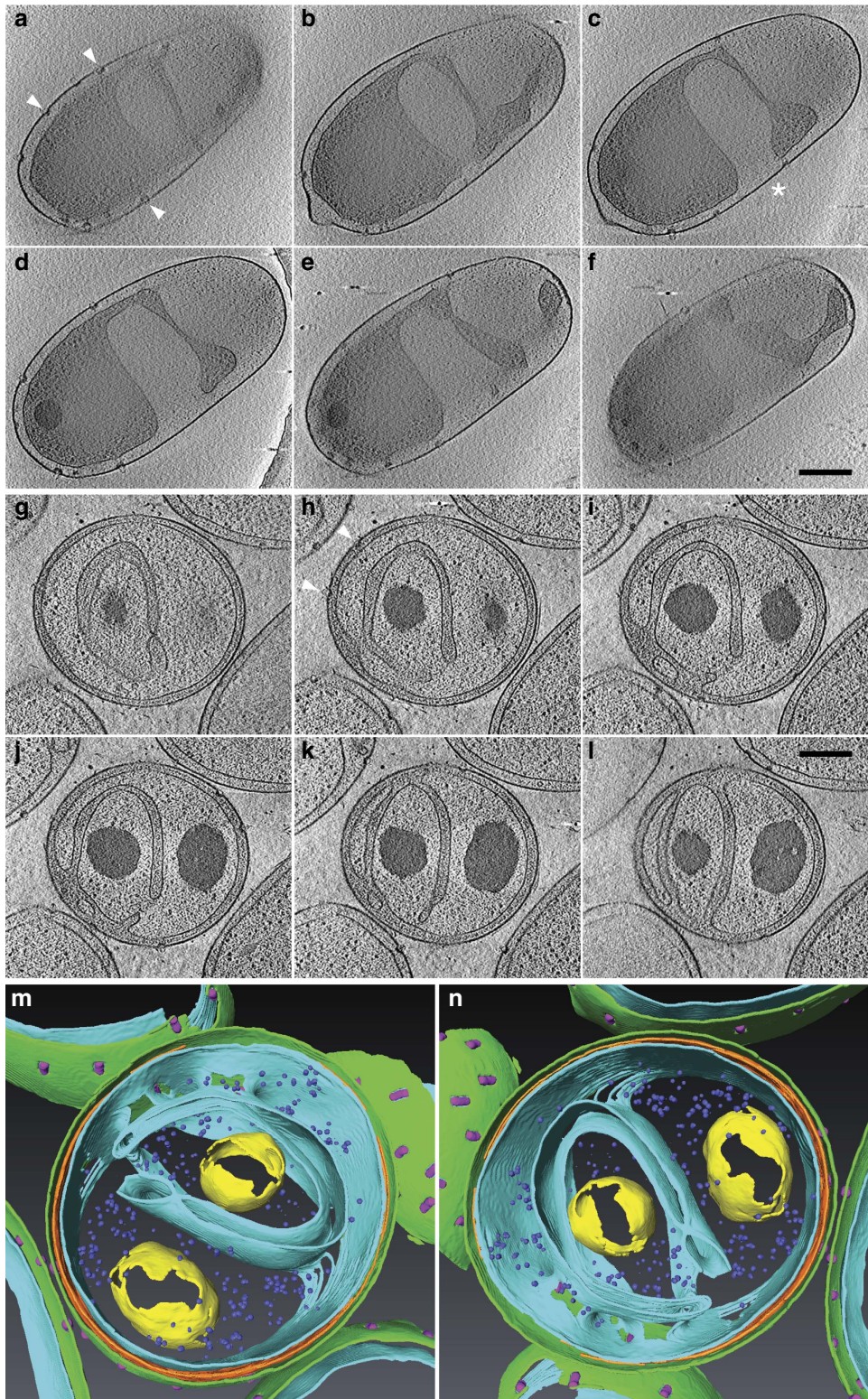

**Figure 3 | Cryo-electron tomography of *Planctopirus limnophila* cells.** (**a–f**) Representative slices of a whole cell tomogram with inter-slice distances of 71, 71, 64, 71 and 64 nm from top left to bottom right (Supplementary Movie 1). The membrane is heavily invaginated and creates a huge periplasmic space. Crateriform structures ('pits') in the cell wall are always in close contact to the cell membrane (arrowheads) and are missing at sites of invagination (asterisk). (**g–l**) Slices of a tomogram of a cell thinned to 320 nm by focused ion beam (FIB) micromachining. Inter-slice distances are 41 nm throughout. Example of a cell with complex membrane invaginations creating a 'channel system' of the periplasm inside the inner cell volume. Scale bars, 250 nm. (**m,n**) Surface-rendered reconstruction of the cell in (**g–l**) visualized in two different angles (**m,n**). Colour code: cell membrane (light blue), PG (orange), outer membrane (green), crateriform structures (purple), granules (yellow), ribosomes (dark blue).

immunogold labelling of high-pressure frozen, freeze substituted slices of *G. obscuriglobus* cells, and found that significantly more gold particles were located at the innermost membrane (40.5%) than close to the outermost membrane (26.5%, *t*-test: $P = 0.0001$) or to arbitrarily distributed gold particles within (21.4%, *t*-test: $P = 0.0001$) or outside of cells (11.6%, *t*-test: $P = 0.0001$) (Fig. 4e, Supplementary Fig. 5c,d and Supplementary Table 1).

In conclusion, although we cannot exclude the presence of ATPases at the outer membrane, our results show that most ATPases are situated at the energized innermost (cytoplasmic) membrane.

**The membrane proteome of *P. limnophila*.** If the innermost membrane is the cytoplasmic membrane, then the outermost membrane should represent a Gram-negative outer membrane (OM, Fig. 1). To test this hypothesis, we analysed the existence of characteristic OM proteins by two different approaches. First we employed bioinformatics to identify homologues of well described OM proteins with a focus on secretion systems in *P. limnophila*, *G. obscuriglobus*, and *R. baltica*. We found many proteins encoded in all three planctomycetal genomes that belong to various secretion pathways (see Supplementary Fig. 6 for summary). While all three genomes encoded nearly all proteins of the Sec secretion pathway, only the genome of *R. baltica* contained a high number of Type VI secretion system-related genes.

In a second approach, we analysed the *P. limnophila* cell envelope proteome. Using the PSORT algorithm, 31 putative outer membrane proteins were predicted to be encoded in the genome, which is similar to the average number predicted for other Gram-negatives (34,8 according to PSORTdb[30]). LC-MS analysis of the *P. limnophila* proteome membrane fraction confirmed 77% of the predicted proteins (24 in total, Supplementary Table 2). 16 of these proteins are related to secretion and transport systems and fit our bioinformatic localization prediction (Supplementary Fig. 6b). In addition, for 11 of these proteins a localization in the outer membrane had been previously predicted by others[21]. In addition to the 24 outer membrane proteins, we detected 24 extracellular proteins, 34 periplasmic proteins, 366 cytoplasmic membrane proteins and 399 proteins of unknown localization within the membrane fraction of the *P. limnophila* proteome. Among proteins with unknown localization, a putative TolC homologue and a previous predicted OMP[21] where identified by manual inspection (ADG67044.1 and ADG67856.1 in Supplementary Table 2). That is, we identified typical Gram-negative cell envelope proteins (Supplementary Table 2).

**Macromolecule uptake in Planctomycetes.** Endocytosis-like uptake is key for the hypothesis that Planctomycetes and eukaryotes share some sort of evolutionary link[6], but it conflicts

WF

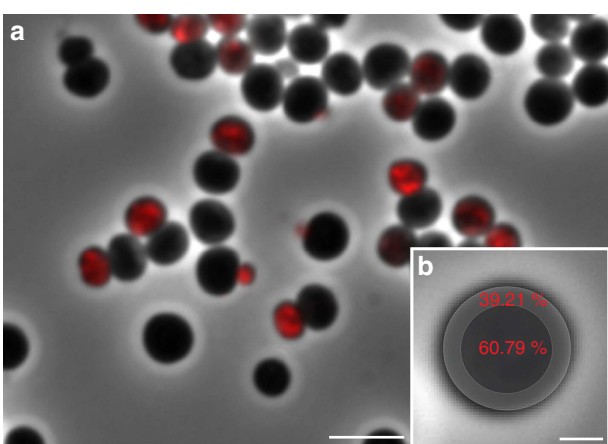

dSTORM

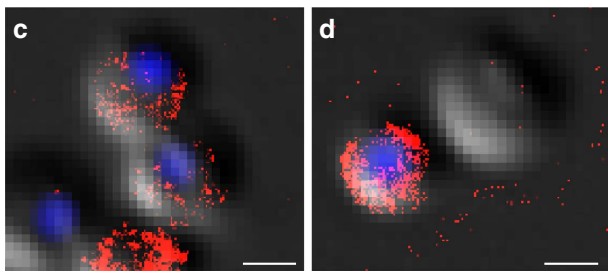

Immunogold

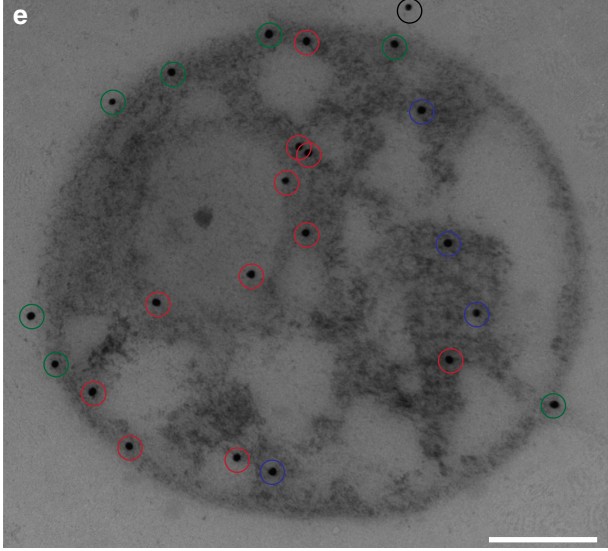

○ Inner membrane   ○ Unspecific in cell
○ Outer membrane   ○ Background

**Figure 4 | The innermost planctomycetal membrane is energized.**
(**a**) Overlay of the wide field (WF) fluorescence signal from a representative ATPase localization experiment (red, anti-Na⁺-F1F0-ATPase antibody) with the corresponding phase contrast (phaco) micrograph of *G. obscuriglobus* cells. Scale bar, 5 µm. (**b**) Among 50 analysed cells in five fields of view, 39.21% of the fluorescence signals were detected in the outer rim, as indicated by the schematic drawing overlaying a phaco micrograph of a single *G. obscuriglobus* cell. 60.79% of the signals were detected within the cytoplasmic area of cells (*t*-test: $P = 0.0001$, see Supplementary Fig. 5a,b for details). Scale bar, 1 µm. (**c,d**) Representative overlays of super-resolution direct stochastic optical reconstruction microscopic (dSTORM) ATPase localization experiments (red, anti-Na⁺-F1F0-ATPase antibody) with the corresponding differential interference contrast micrograph of *G. obscuriglobus* cells. Scale bars, 1 µm. (**e**) Immunogold-labelling of the ATPase (black dots, anti-Na⁺-F1F0-ATPase antibody) in high pressure-frozen and freeze-substituted *G. obscuriglobus* cells (see Supplementary Fig. 5c,d for detail). The micrograph shows four different localizations of gold particles which are either associated with the inner membrane (red circles), the outer membrane (green circles) or show non-specific localizations either within the cell (blue circles) or outside of the cell (black circles). Scale bar, 0.2 µm.

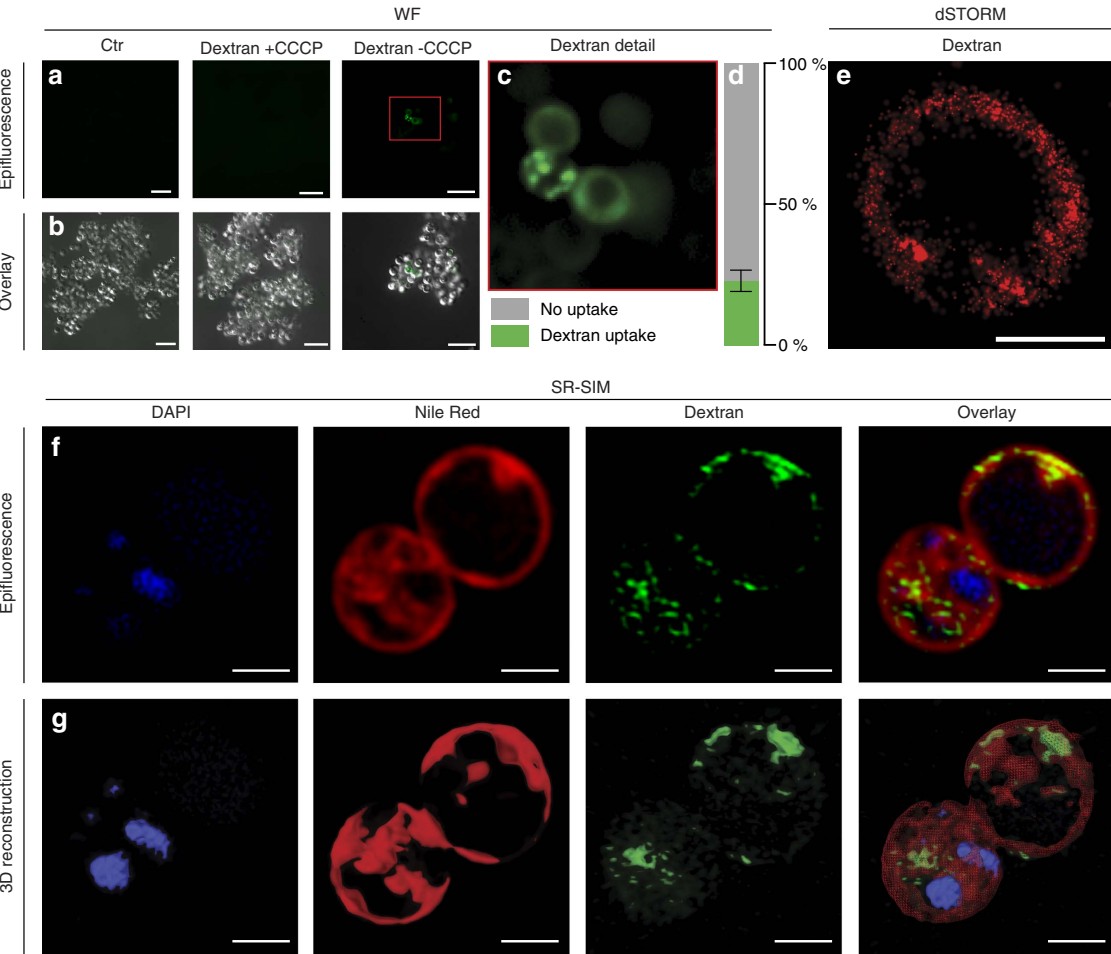

**Figure 5 | *Gemmata obscuriglobus* takes up high-molecular weight dextran polysaccharides. (a–c)** Wide field (WF) microscopy of *G. obscuriglobus* cells that were either untreated (Ctr) or fed with fluorescein-labelled 40 kDa dextran (Dextran). Prior to addition of dextran cell samples were either poisoned with carbonyl cyanide m-chlorophenyl hydrazon (+CCCP) or left untreated (−CCCP). Epifluorescence overview micrographs (**a**) were combined with differential interference contrast images of the same field of view (**b**). Dextran fluorescence signals (green) were exclusively detected in −CCCP cells, indicating an active uptake. Scale bars, 10 µm. (**c**) Magnification of the red framed section in **a** shows distinct foci of dextran accumulation. Scale bar, 10 µm. (**d**) While representative images are shown in (**a–c**), in total 1,065 cells among 7 fields of view were analysed and 23% of them showed dextran uptake (error bars indicate s.d.). In contrast, among 816 CCCP treated cells not a single uptake event was observed. (**e**) Super-resolution direct stochastic optical reconstruction microscopic (dSTORM) analysis of *G. obscuriglobus* cells fed with Alexa 467-labelled 10 kDa dextran. The red fluorescence signals show an almost homogeneous distribution of dextran in some areas of the cell (see Supplementary Fig. 8a–f for details). Scale bar, 1 µm. (**f**) Super-resolution structured illumination microscopy (SR-SIM) of *G. obscuriglobus* cells with DAPI stained nucleoids (blue) and Nile Red stained membranes fed with fluorescein-labelled 40 kDa dextran (green). The nucleoid is highly condensed and multiple invaginations of the innermost membrane are visible that colocalise with the invaginated innermost membrane (Overlay). 3D reconstruction with Amira 6.0 shows dextran within the *G. obscuriglobus cell* (**g** and Supplementary Movie 3). Scale bars, 1 µm.

with the existence of a typical bacterial outer membrane. We therefore revisited previously performed GFP uptake experiments with *G. obscuriglobus*[16]. In 10.15% of the 1,281 analysed cells, we observed GFP uptake and the formation of GFP-containing foci in WF fluorescence microscopic experiments. Treatment of the cells with CCCP prevented uptake (0% uptake among 1,294 analysed cells), indicating an active process (Supplementary Fig. 7a–d) as previously described[16]. However, when *G. obscuriglobus* cells were analysed with super-resolution structured illumination microscopy (SR-SIM), the GFP signals were located within the invaginations of the cytoplasmic membrane (Supplementary Fig. 7h). Three-dimensional recons-tructions illustrated the internalization of GFP into *G. obscuriglobus* cells, and the overlay with stained membranes (Nile Red) supported the localization of GFP molecules in the enlarged periplasmic space (Supplementary Fig. 7i and

Supplementary Movie 2). Applying corresponding experimental settings, we did not observe GFP internalization by *P. limnophila* cells (Supplementary Fig. 7e–g).

Since GFP is an artificial substrate for Planctomycetes, we then looked for a more physiologically relevant experimental setting. Planctomycetes can utilize various carbon compounds[2], and we recently found that they can live on high molecular weight dextran as sole carbon source[3]. We fed *G. obscuriglobus* cells with 30 kDa-sized fluorescently labelled dextran and observed the same distinct foci as in the GFP feeding experiments within 23% of the 1,065 analysed cells that took up the dextran (versus none of the 816 analysed CCCP-treated cells, Fig. 5a–d). But again, super resolution microscopy (dSTORM) revealed an accumulation of dextran (10 kDa Alexa 647) in the periplasmic space only (Fig. 5e). Cluster analysis of dSTORM images revealed that dextran did not accumulate in 50–200 nm sized vesicles,

but rather is distributed in the periplasmic space (Supplementary Fig. 8). This finding was further corroborated by SR-SIM microscopy which allowed parallel staining of membranes and DNA (Fig. 5f). Three-dimensional reconstructions gave analogous results compared to GFP uptake experiments (Fig. 5g and Supplementary Movie 3). Most importantly, with dextran as substrate, similar results were obtained also with *P. limnophila,* where 22.17% of the 15,557 analysed cells took up the substrate (Supplementary Fig. 9 and Supplementary Movie 4).

Since genetic tools exist for *P. limnophila*[25] but not for *G. obscuriglobus*, we explored the potential role of membrane coat-like proteins (MCs), expected to function in planctomycetal endocytosis, in *P. limnophila*. First, we used the MC-like protein gp4978 suggested to play a key role in GFP uptake in *G. obscuriglobus* to detect its homologue in *P. limnophila*. In addition to classical sequence-based comparison, we employed structure prediction as previously used to identify MCs in Planctomycetes[15] (Supplementary Fig. 10). While structure-based analysis revealed ten putative MC-like proteins, only one of them, Plim_1972, comprised high sequence similarity to the *G. obscuriglobus* MC-like protein (40% identity versus 29%-22% for other putative MCs) above thresholds previously used to determine homology among planctomycetal proteins[22].

Plim_1972 was subsequently deleted from the *P. limnophila* genome, resulting in the *P. limnophila* ΔMC mutant. The deletion was verified through PCR experiments targeting gDNA of both mutant and wild type (WT) cultures (Supplementary Fig. 11a,b). Furthermore, we used synthesized oligopeptides to produce an antibody against Plim_1972. Western blot analysis of the WT in comparison to the mutant revealed that the protein was absent in the latter (Supplementary Fig. 11c). The *P. limnophila* ΔMC mutant was then used in dextran feeding experiments. In three independent experiments with three technical replicates each, no significant differences were detected in the uptake of labelled dextran between the mutant and the WT after analysing 15,556 and 19,502 cells respectively (t-test: $P = 0.21$, Supplementary Fig. 11d–f). Thus, our results demonstrate that Plim_1972 is not required for dextran internalization. However, we cannot exclude the possibility that other MC-like proteins may be involved in this process.

**Planctomycetal appendages**. To study how dextran passed through the outer membrane of Planctomycetes, we analysed the surface of *P. limnophila* and *G. obscuriglobus* by scanning electron microscopy (SEM) and transmission electron microscopy (TEM). We found that the outer membrane of *P. limnophila* displays two different types of crateriform structures. Large crateriform structures were distributed all over the cell surface (Fig. 6a and Supplementary Fig. 3), with some restrictions (Fig. 3 and Supplementary Movie 1). In some cases, pili-like fibres with a diameter of 12 nm were attached to the large crateriform structures (Fig. 6b). In contrast, small crateriform structures were exclusively found at the pole and were associated with the holdfast-structure and the stalk, which measure 6 nm in diameter[25] (Fig. 6c). Light microscopic analysis allowed visualization of both types of fibres, the stalk and the pili-like in *P. limnophila* and *G. obscuriglobus* (Fig. 6d,g). Feeding experiments with gold-labelled dextran molecules revealed gold particles primarily at pili-like fibres associated with the large crateriform structures of *P. limnophila* and *G. obscuriglobus* (Fig. 6e,f,h,i). Some pili-like fibre bundles were longer than the entire cell body (Fig. 6j). Some gold particles appeared to be internalized in such experiments (Fig. 6j). Thus, these results indicate that dextran binds to fibres that originate from large crateriform structures (Fig. 6k).

## Discussion

Planctomycetes have challenged our concepts of microbial structure and function because of unusual characteristics such as apparent cytosol compartmentalization and ability to perform endocytosis-like macromolecule uptake (Fig. 1; for review see ref. 1). While recent work challenged this view[20–24], producing definitive evidence has been hampered by the paucity of genetic tools[31]. In this study, we developed methods for *gfp* expression and gene deletion in a model planctomycete to study its cell plan and macromolecule uptake.

Construction of a GFP-expressing *P. limnophila* strain in combination with membrane staining and (super-resolution) microscopy revealed that the innermost membrane possesses features of a cytoplasmic membrane. In contrast to other bacteria, the planctomycetal cytoplasmic membrane can comprise multiple invaginations, leading to enlargements of the periplasmic space (Fig. 2). Plasmolysis experiments demonstrated that such enlargements can be further expanded (Supplementary Fig. 1). Freeze-fracturing (Supplementary Fig. 2h,i) and cryo-electron tomography (CET) of frozen-hydrated *P. limnophila* cells (Fig. 3 and Supplementary Fig. 3) showed a typically Gram-negative cell envelope consisting of an outer membrane, a peptidoglycan layer and a cytoplasmic membrane[22]. Furthermore, tomographic reconstructions revealed that the membrane invaginations are interconnected (Fig. 3, Supplementary Fig. 3 and Supplementary Movie 1). $DiOC_6(3)$ staining supports that the innermost cytoplasmic membrane is energized (Supplementary Fig. 4). Similar results have been previously obtained for *G. obscuriglobus*, but the ability of $DiOC_6(3)$ to visualize the membrane potential was not acknowledged[32]. Our results are consistent with the ATPase localization at the innermost membrane of *G. obscuriglobus* (Fig. 4). Previous studies on anammox Planctomycetes proposed a different ATPase localization, either on all three membranes[33], or only on the anammoxosome membrane[34]. However, this might have been caused by 'moonlighting' of F1-ATPase domains[35,36], or may be explained by effects of specific anammox structures leading to mislabelling[37]. Our microscopic observations are consistent with bioinformatic predictions of the outer membrane secretion systems and with the analyses of the *P. limnophila* membrane proteome (Supplementary Fig. 6). Both methods revealed proteins typical for the outer membrane of Gram-negative bacteria such as porins, as well as typical periplasmic proteins and proteins that are associated with the cytoplasmic membrane (Supplementary Table 2). Our findings are also in line with the recent report of lipid A in Planctomycetes[24], a key component of lipopolysaccharides in outer membranes.

Together, our results support a Gram-negative cell plan for planctomycetes, with a notable difference. In contrast to most Gram-negative bacteria, the planctomycetal periplasm appears to be variable among individual cells, sometimes leading to great enlargements of the periplasmic space caused by invaginations of the cytoplasmic membrane. Interestingly, such membrane invaginations have been previously reported in pathogenic Chlamydia, which belong to the same Planctomycetes-Verruco-microbia-Chlamydiae superphylum[38].

A planctomycetal Gram-negative cell plan challenges the existence of an endocytosis-like process[16]. While Gram-negative bacteria frequently produce outer membrane vesicles[39], invagination of their outer membrane and formation of periplasmic vesicles is hindered by its asymmetric lipid architecture and the peptidoglycan cell wall[21,22]. Furthermore, the periplasmic space is devoid of high-energy molecules such as ATP or GTP[40] that are required for clathrin-mediated endocytosis in eukaryotes[41]. Thus, the formation of endocytic vesicles by MC-proteins at the outer membrane of

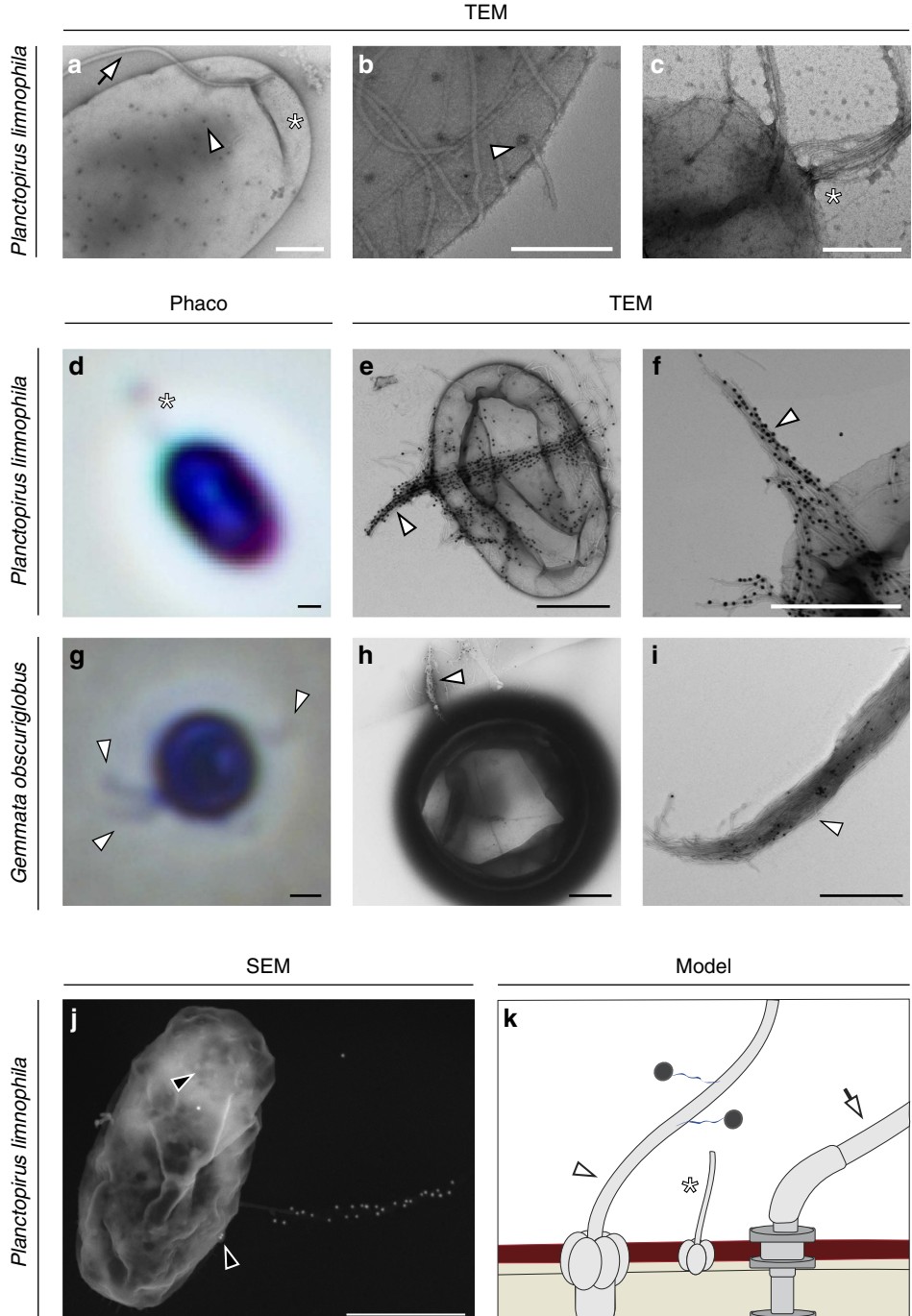

**Figure 6 | *Planctopirus limnophila* and *Gemmata obscuriglobus* possess fibres that bind to dextran.** (**a–c**) TEM analysis of *P. limnophila* cells revealed multiple distinct appendages. The flagellum (**a**, white arrow) with a diameter of 22 nm is typical for bacteria. In contrast, large- (**a,b**, white arrowheads) and small crateriform structures (**a,c**, asterisk) are unique to the Planctomycetes and form pili with diameters of 12 nm (**b**) 6 nm (**c**) respectively. The polar thinner fibres (**a,c**, asterisks) form a stalk that attaches the cell to a surface or is involved in cell aggregation. Scale bars, 0.2 μm. (**d–i**) Light microscopic phase contrast (**d,g**, Phaco) and TEM (**e,f,h,i**) micrographs of *P. limnophila* (**d–f**) and *G. obscuriglobus* cells (**g–i**) fed with gold-labelled dextran (four individual experiments). In the light microscope (**d,g**) either the stalk of *P. limnophila* (asterisk), that is associated with the small crateriform structures (**c**), or the fibres of *G. obscuriglobus* associated with the large crateriform structures (**b**, white arrowheads) are visible. TEM analysis revealed dextran binding to fibres of both species (**e,f,h,i**). (**j**) Backscattered electron detection in the SEM micrograph shows electron dense gold particles as bright dots and illustrates that fibre bundles can even be longer than the entire cell body. Some gold particles appear to be internalized by the cell (**j**, black arrowheads). The scheme (**k**) shows the flagella/attachment pole of a *P. limnophila* cell, which is opposite to the reproduction pole where cell division through budding takes place. While small crateriform structures are localized only close to the flagellum and produce the stalk (asterisk), fibres associated with the large crateriform structures are distributed throughout the cell (white arrowheads). Scale bars, 0.5 μm.

Planctomycetes appears puzzling. We revisited *G. obscuriglobus* GFP uptake experiments[16] and obtained similar results using WF microscopy (Supplementary Fig. 7a–d). However, when GFP uptake was analysed with SR-SIM, we found GFP in close proximity to invaginations of the cytoplasmic membrane (Supplementary Fig. 7h,i and Supplementary Movie 2). Similar results were obtained for the active uptake of fluorescently labelled dextran[42] in *G. obscuriglobus*, and *P. limnophila* (Fig. 5; Supplementary Fig. 8). Super resolution microscopy suggested that dextran is homogeneously stored in the enlarged periplasm and not in large endocytic vesicles as previously proposed[16] (Fig. 5, Supplementary Figs 8 and 11, Supplementary Movies 3 and 4). Comprehensive CET analysis did not reveal any of the previously proposed 50–200 nm endocytic vesicles in *P. limnophila*, but demonstrated continuity of the innermost cytoplasmic membrane (Fig. 3, Supplementary Fig. 3 and Supplementary Movie 1); the innermost membrane was sometimes heavily invaginated, which might be misinterpreted as vesicles. These observations are in line with the finding that the uptake of macromolecules was not affected in the *P. limnophila* ΔMC mutant compared to *P. limnophila* wild type cells. This mutant lacks a membrane coat-like protein that is similar to the one previously proposed to localize at endocytic vesicles in *G. obscuriglobus*[16]. However, since *P. limnophila* possesses more than one MC-like gene, we cannot exclude that other MC-like proteins may be involved in macromolecule uptake. Based on our results and our interpretation of previous work, we conclude that Planctomycetes do not seem to use a vesicle-mediated mechanism for macromolecule uptake into the periplasm.

Nevertheless, both dextran and GFP must somehow cross the planctomycetal outer membrane to accumulate in the enlarged periplasm. Nonspecific pores, formed by typical outer membrane porins (similar to the ones identified in our proteomic approach, Supplementary Table 2), can facilitate uptake of ions and small molecules up to a size of about 600 Da but not of macromolecules (for review see ref. 43). Thus, transfer of GFP or dextran via these porins would require extracellular degradation. However, degradation would destroy the GFP fluorescence. Consequently, GFP (and likely high-molecular weight dextran) may be internalized by a different mechanism. We speculate that this process might be associated with the crateriform structures, unique planctomycetal outer membrane complexes and their associated fibres. Our CET-analyses revealed that the crateriform structures appear to form press stud-like connections between the outer membrane and the cytoplasmic membrane; these connection sites are different from other zones of membrane adhesion previously reported for Gram-negative bacteria[44].

In summary, our findings further support the idea that the planctomycetal cell architecture is Gram-negative. Yet, Planctomycetes remain peculiar, with complex cytoplasmic membrane invaginations, enigmatic crateriform structures and an unusual macromolecule uptake system that has still to be characterized.

*Note added in proof.* A recent publication reports the presence of nuclear pore-like structures in the planctomycete *Gemmata obscuriglobus*[45].

## Methods

**Cultivation conditions.** *Planctopirus limnophila* DSM 3776 (the bacterium formerly known as *Planctomyces limnophilus*) and *Gemmata obscuriglobus* DSM 5831 were cultivated at 28 °C in limnic medium 3 (M3) composed of $1 g l^{-1}$ peptone, $1 g l^{-1}$ yeast extract, $1 g l^{-1}$ glucose, 5 ml vitamin solution (double concentrated) and $20 ml l^{-1}$ mineral salt solution buffered with 10 mM HEPES at pH 7.5. *Rhodopirellula baltica* SH1 DSM 10527 was cultivated at 28 °C in marine medium 2 (M2) composed of $1 g l^{-1}$ peptone, $1 g l^{-1}$ glucose, 10 ml vitamin solution (double concentrated), 250 ml double-concentrated artificial sea water ($46.94 g l^{-1}$ NaCl, $7.84 g l^{-1}$ Na$_2$SO$_4$, $21.28 g l^{-1}$ MgCl$_2$ × 6H$_2$O, $2.86 g l^{-1}$

CaCl$_2$ × 2H$_2$O, $0.384 g l^{-1}$ NaHCO$_3$, $1.384 g l^{-1}$ KCl, $0.192 g l^{-1}$ KBr, $0.052 g l^{-1}$ H$_3$BO$_3$, $0.08 g l^{-1}$ SrCl$_2$ × 6H$_2$O, $0.006 g l^{-1}$ NaF) and $20 ml l^{-1}$ mineral salt solution buffered with 5 mM Tris/HCl at pH 7.5. *Escherichia coli* K12 Top10, was cultured at 37 °C in lysogeny broth composed of $5 g l^{-1}$ yeast extract, $10 g l^{-1}$ trypton and $10 g l^{-1}$ NaCl at 7.0 pH.

**Construction of a GFP *Planctopirus limnophila* strain.** The constitutive GFP expression system consists of a TN5 transposon, harbouring the GFPmut2 (ref. 46) gene along with a kanamycin resistance cassette under the transcriptional control of the GAPDH promoter from *P. limnophila*. Briefly, the construction was based on the pMOD3 <R6Kγ ori/MCS> vector (Epicentre). This pMOD3 vector and the pKen GFP mut2 plasmid (Addgene) were both cut with *Xma*I and *Pst*I (Fermentas). Vector backbone and the GFP insert were ligated (NEB quick ligation) to construct pCJ0001. This plasmid and the PCR product of plasmid pCR2.1 (Invitrogen) (primer CJ326: 5′-GTC AAT CGA TGC GGT TTT ATG GAC AGC AAG-3′ and CJ327: 5′-AGC TGA ATT CGC GAC ACG GAA ATG TTG AAT-3′) were cut with *Cla*I and *Eco*RI. Subsequently, the pCJ0001 backbone was ligated to the kanamycin resistance cassette obtained from pCR2.1 to construct pCJ0002. Then, the GAPDH promoter was amplified (Phusion DNA polymerase, NEB) from *P. limnophila* gDNA (primer CJ403: 5′-AGC GAG AGA ATC AGG CTT ACC-3′ and CJ404: 5′-AGC CAA CGT TTC ATG CAT ATC-3′). For further processing, a subsequent PCR was performed on the amplified product to attach *Xma*I and *Eco*RI restriction sides (Primer CJ340: 5′-ACT GAA TTC ATC ACC TGT TGA GGC GAT TC-3′ and CJ341: 5′-ACT CCC GGG CAA CTG AGA TAC AGT CTG GA-3′). PCR product and pCJ0002 were cut with *Xma*I and *Eco*RI and ligated to yield pCJ0003. pCJ0003 was used to build the Tn5 transposon according to the epicentre PCR protocol (Epicentre Cat. No. MOD1503). Transposons were transferred into *P. limnophila* and insertion sites of ten clones were determined by arbitrary PCR as previously described[25]. Two clones (SCJ0034-1.2 and SCJ0036-2.2) showed similar cytosolic GFP localization and identical growth characteristics compared to the *P. limnophila* wild type. Strain SCJ0036-2.2 was used for this study.

**Staining of planctomycetal cells.** Membranes of *P. limnophila*, *R. baltica*, *G. obscuriglobus* and *E. coli* cells were stained with (N-(3-Triethylammonium-propyl)-4-(6-(4-(Diethylamino) Phenyl) Hexatrienyl) Pyridinium Dibromide (FM4–64/FM4-64FX)[25] (ThermoFisher) or Nile Red (Sigma-Aldrich) at a final concentration of $3 \mu g ml^{-1}$. Samples were incubated for 10 min at room temperature (RT). In addition, DAPI was added at a final concentration of $1 \mu g ml^{-1}$ for 10 min at RT. Cells were washed twice with 1 ml tap water and centrifuged at 2,500 × g for 1.5 min. For DiOC$_6$(3) staining of *P. limnophila*, *R. baltica* and *G. obscuriglobus* cells were incubated for 45 min at 28 °C at a final concentration of $5 \mu g ml^{-1}$ DiOC$_6$(3) (ThermoFisher). To induce plasmolysis of *P. limnophila* and *E. coli*, cells were treated as previously described[28]. In brief: sucrose concentrations of 30% were applied for 3 min at RT respectively. Subsequently, cells were fixed with 1% glutaraldehyde for 1 h at RT. Afterwards, cells were stained with Fm4-64FX and DAPI as described above. For quantification, we used NIS elements 4.2 and 4.3 (Nikon) to determine signal intensity plots or intensity per area measurements. Signal intensities of line plots were consistently adapted due to different staining efficiencies. Cytosol and nucleoid size were measured using the GFP and DAPI signal respectively in comparison to the whole cell size determined in phase contrast applying the Auto Detect ROI analysis feature. NIS elements 4.3 were used to determine co-localization. For *P. limnophila* and *R. baltica*, 50 individual cells each were compared. Due to high-aggregation of *G. obscuriglobus* cells entire fields of view were analysed.

**Wide field fluorescence microscopy.** For sample immobilization, MatTek Glass Bottom Microwell Dishes (35 mm dish, 14 mm microwell with No. 1.5 cover-glass P35G-1.5-14-C) were used. To minimize drift and cell movement, 1% agarose pads were placed on top of 3 μl samples. To prevent evaporation, but to achieve optimal phase contrast imaging, the plastic lid was removed and the agarose pads were covered with an additional high-precision coverslip (LH24.1 Carl Roth GmbH). The coverslip was sealed against the plastic dish with grease (Vaseline, Lenhart Kosmetik). WF fluorescence- and Phaco samples were visualized on a Nikon Eclipse Ti inverse microscope with DAPI (370/36–440/40), GFP (485/20-525/30) and Fm4–64 (525/30-705/72) filters. Fluorescence z-stacks and bright-field images were taken using a Nikon N Plan Apochromat λ ×100/1.45 oil objective and the ORCA FLASH 4.0 HAMMATSU or Nikon DS-Ri2 cameras, respectively. Images were processed using the NIS-elements imaging software V4.2 and V4.3 (Nikon) together with the 3D Landweber Deconvolution algorithm (Z-step: 0.2 μm, spherical aberration: 0.2).

**Super resolution structured illumination microscopy (SR-SIM).** Samples were visualized on a Zeiss LSM 780 with ELYRA PS.1 (Carl Zeiss AG) with 561, 488 and 405 nm lasers and BP 570–650 + LP 750, BP 495–575 + LP 750 and BP 420–480 + LP 750 beam splitters. Z-Stack images were taken using Plan-Apochromat ×63/1.4 oil DIC M27 objective and processed using the software ZEN2011 (Carl Zeiss AG). Images were post processed using the Amira 6.0 3D image analysis software (FEI). To compare WF and SR-SIM

resolution in biological samples, intensity maxima of Fm4–64 or Nile-Red stained membranes of 20 cells were compared using WF (Nikon Eclipse Ti) and SR-SIM (Zeiss LSM 780), respectively. In addition, a nanoruler GATTA-SIM120B (GATTAquant), with a defined distance of 120 nm of two Alexa Fluor 488 molecules, was used to determine the resolution of the Zeiss LSM 780.

**Cryo-electron tomography of _P. limnophila_ cells.** A late-exponential-phase culture of _P. limnophila_ was gently filtered (10 µm membrane filter, Whatman Nuclepore) to remove aggregated cells. Aliquots (3 µl) of the filtered cell suspension were mixed with the same volume of BSA-stabilized 15 nm colloidal gold solution (Aurion), and placed on holey carbon-coated 200 mesh copper grids (R2/1, Quantifoil, Jena, Germany) immediately before thin-film vitrification by plunge-freezing in liquid propane (63%)/ethane (37%) (ref. 47). Typically, grids with frozen-hydrated samples were mounted in Autogrids[48] and ~200 nm thin lamellae of vitrified material were milled with 30 keV gallium ions after application of a protective platinum layer in a dual-beam (FIB/SEM) instrument (Quanta 3D FEG, FEI, Hillsboro, OR, USA) equipped with a Quorum cryo-stage maintained at −185 °C (PP2000T, Quorum, East Sussex, UK). Milling was carried out at nominal incident ion beam angles of 16° to 20° (9° to 13° effectively) using gallium beam currents of 300, 100 and 30 pA in sequential milling steps[49]. Afterwards tomographic tilt series were recorded under low dose conditions (total dose typically 150 e Å$^{-2}$) on a Tecnai G2 Polara (FEI, Eindhoven, the Netherlands) equipped with a post-column energy filter and a 2 k CCD camera (MultiScan) or a K2 summit direct electron detector (Gatan, Pleasanton, CA, USA). For tilt series recorded with the direct detection device, dose fractionation mode was employed and subframes of each projection were sampled, which were then aligned to compensate for beam-induced object drift, using an in-house implementation of the algorithm from the study by Li _et al._[50]

Typically, tilt series were recorded at a nominal defocus of −5 or −6 µm, and a primary magnification of ×27,500 (corresponding to pixel sizes on the object level of 0.427 nm (K2) and 0.805 nm (2 k CCD)), and covered an angular range of ±60° in increments of 1.5° or 2°, respectively. IMOD v4.7.8 (ref. 51) was used for 3D reconstruction, and MatLab8 (MathWorks) incorporating the TOM toolbox[52] for all image processing. Segmentation of three times binned volumes was done in Amira v6.0.1 (FEI, Eindhoven, the Netherlands) with specific automatic membrane segmentation[53].

**Freeze-etching.** Freeze-etching was performed on cells taken from a _P. limnophila_ culture at two different time points. The cells were concentrated (centrifugation 800g for 4 min) and inserted either in a gold dome-shaped carrier (1.7 µl per carrier) or in a jet-freeze carrier sandwich with a grid in between (3 µl for the total sandwich). The carriers were plunge frozen in liquid nitrogen. The frozen samples were introduced into a Balzers BAF400 freeze-etch machine precooled to −150 °C with pressure below 10$^{-7}$ bar. The samples were kept at −97 °C for 7 min before being fractured, after they were allowed to freeze-etch (sublimation of water) for 4 min. The samples were shadowed with 1 nm Pt-C (angle 45°) and 10 nm C (angle 90°). Subsequently, biological material was removed from the replicas by overnight incubation in 70% sulfuric acid. The replicas were washed twice on ddH$_2$O and picked up with 700-mesh hexagonal copper grids. The grids were investigated at 60 kV in a JEOL JEM-1010 TEM instrument.

**Immunofluorescence microscopy.** For immunofluorescence microscopy 2 ml of an exponential-phase _G. obscuriglobus_ culture was centrifuged at 2,500g for 2 min. The supernatant was discarded and cells were dissolved in 3% paraformaldehyde (PFA). After incubation at 4 °C over night (ON) cells were washed three times with phosphate buffered saline (PBS). For permeabilisation, cells were resuspended in 150 µl ice cold resuspension buffer (50 mM Tris/HCL, 10 mM Na$_2$EDTA X 2H$_2$O pH 8.0, 0.1 mg ml$^{-1}$) and subsequently 150 µl disruption buffer (200 mM NaOH, 1% SDS) was added. Samples were five times gently inverted and incubated for 15 min at RT. Afterwards, cells were washed three times in PBS and for further treatment dissolved in 1 ml ×1 blocking reagent (Roche, Germany) and incubated for 2 h at room temperature (RT) with agitation (300 r.p.m.). The primary antibody (anti-Na$^+$-F$_1$F$_0$-ATPase[54]) diluted 1:100 in ×1 blocking reagent. Samples were incubated over night at 4 °C, washed three times with PBS and treated with the secondary antibody (Alexa-Fluor 488 goat anti-rabbit) 1:1,000 diluted in PBST and incubated for 3 h at RT in the dark. Finally, cells were washed three times with PBST and diluted in a corresponding volume of mounting medium (20 mM Tris pH 8.0, 0.5% N-propyl gallate, 90% glycerol) for WF fluorescence microscopy. For dSTORM experiments cells were finally diluted in dSTORM-MEA-buffer[55]. For WF quantification, 50 positive labelled cells were divided into two parts; the outer rim (400 nm) and the inner part. The intensity per area was determined for each part using NIS elements 4.11 ROI analysis function.

**Immunolabelling on ultrathin sections.** Samples were high pressure frozen (Leica EM Pact) and freeze substituted with a 98% ethanol solution containing 2% water and 0.5% formaldehyde (−90 °C for 50 h, temperature rise 10° in 4 h; −50 °C for 24 h, temperature rise 10° in 4 h; −30 °C for 12 h). Samples were transferred into 100% ethanol at −30 °C with several changes and then allowed to reach a temperature of 7 °C and embedded in LRWhite resin. After polymerization,

ultrathin sections were cut with a diamond knife, collected with butvar-coated nickel grids and incubated with the anti-Na$^+$-F$_1$F$_0$-ATPase[54] IgG antibody (1:20 dilution) overnight at 7 °C. After washing with PBS, sections were placed onto drops of protein A-gold (1:75 dilution, 15 nm in size) and were incubated for 1 h at room temperature. After washing with PBS containing 0.1% Tween 20, sections were further washed with TE- buffer and distilled water. Counterstaining of the sections was performed with 4% uranyl acetate for 1 min before examination in a Zeiss TEM 910 transmission electron microscope. For quantification 10 micrographs (3.5 µm$^2$) were analysed independently by three different researchers.

**Secretion system proteins in Planctomycetes.** Essential secretion system proteins (secretion system I to VI, as defined by the KEGG database) were analysed with BLASTp and PSI-BLAST in the NCBI database against _Planctopirus limnophila_, _Gemmata obscuriglobus_ and _Rhodopirellula baltica_ genomes. Proteins with a higher identity than 30%, an e-value lower than 1e-6 and with conserved domain architecture were assumed as correct. Only proteins with positive reciprocal blast verification are listed. If no homologue was found boxes are marked in white (Supplementary Fig. 6). In cases the protein sequence was notified by the KEGG database for one of the three analysed Planctomycetes this sequence was used as reference sequence for further analyses.

**Proteome analysis.** 500 ml of a _P. limnophila_ culture, cultivated for 3 days at 28 °C and 90 r.p.m. was harvested by centrifugation (5,000 × g for 15 min). The pellet was resuspended in 30 ml ice cold 20 mM MOPS buffer pH 8, 200 mM NaCl containing one dissolved EDTA-free protease inhibitor tablet (Roche). The cells were disrupted by cell homogenization (3 × 30 s 6.0 m s$^{-1}$; MP Biomedicals). Cellular debris was removed by centrifugation (5,000 × g for 20 min). Membranes were separated from the soluble proteins by centrifugation (160,000 × g, 1 h, 4 °C). To isolate crude membrane fractions containing both cytoplasmic and outer membranes, cell extracts were subjected to isopycnic sucrose gradient centrifugation (1, 1.5, 2, and 2.5 M, 200,000 × g, 24 h, 12 °C). Afterwards, membranes were treated with carbonate buffer to remove membrane associated and cytosolic proteins[56–58] followed by a precipitation step to enrich the membrane proteins[58,59]. Precipitated proteins were washed in 400 µl of icecold methanol and pelleted by centrifugation (13,000 × g, 15 min, 4 °C). The pellet was initially dried in a Speedvac (Eppendorf concentrator plus) for 8 min, air-dried in an extractor hood and finally dissolved in 8 M urea, 2 M thiourea.

Aliquots of 12–30 µg protein were separated via one dimensional SDS–polyacrylamide gel electrophoresis[60]. In gel digestion of proteins was carried out as described[60] by dividing each lane into eight subsamples with similar protein amounts which were densitometrically determined using AIDA software (Raytest Isotopenmeßgeräte GmbH). Extraction and desalting of the resulting peptides was done according to ref. 61.

For liquid chromatography—tandem mass spectrometry (LC–MS/MS) analysis a nanoAQUITY Ultra Performance Liquid Chromatography System (Waters Corporation, Milford, MA, USA) was coupled to an LTQ Orbitrap Velos Pro mass spectrometer (Thermo Fisher Scientific Inc). Peptides from each gel piece were solved in 3% acetonitrile and 0.1% formic acid, centrifuged for 20 min at 109,000 × g and loaded onto a BEH C18 column, 130 Å, 1.7 µm, 75 µm × 250 mm at a flow rate of 0.35 µl min$^{-1}$ (Waters Corporation). Elution of peptides from the column was performed using a 176 min gradient starting with 3.7% buffer B (80% acetonitrile and 0.1% formic acid) and 96.3% buffer A (0.1% formic acid in Ultra-LC–MS-water): 0–30 min 3.7% B; 30–51 min 3.7–22.1% B; 51–61 min 22.1–27.0% B; 61–127 min 27–48.3% B; 127–150 min 48.3–62.5% B; 150–163 min 62.5–99% B; 163–166 min 99% B; 166–171 min 99–3.7% B, 171–176 min 3.7% B.

Primary MS scans were performed in the Fourier transformation mode scanning an m/z of 400–2,000 with a resolution (full width at half maximum) of 60,000 and a lock mass of 445.12003. Primary ions were fragmented in a data-dependent collision induced dissociation mode for the 20 most abundant precursor ions with an exclusion time of 12 s and analysed by the LTQ ion trap. The following ionization parameters were applied: normalized collision energy: 35, activation Q: 0.25, activation time: 10 ms, isolation width: 2 m/z, charge state: +2 to +4. The signal to noise threshold was set to 2,000.

MS/MS data were analysed using MaxQuant (Max Planck Institute of Biochemistry, www.maxquant.org, version 1.4.1.2) and the following parameters: peptide tolerance 5 p.p.m.; a tolerance for fragment ions of 0.6 Da; variable modification: methionine oxidation, fixed modification: carbamidomethylation; a maximum of three modifications per peptide was allowed; a minimum of two unique peptides per protein; fixed false discovery rate of 1%. All samples were searched against a database containing all protein sequences of _P. limnophila_ DSM 3776 extracted from NCBI at 07/21/2015 with a decoy mode of reverted sequences and common contaminants supplied by MaxQuant. A protein was considerably reliable identified when it was identified in two biological replicates.

Detected proteins were analysed using PSI-BLAST in the NCBI database, against NR, expect the taxa Planctomycetes (taxid: 203,682). Data were analysed with an expected threshold of 0.001 and five iterations. For all BLAST searches, an e-value of 1e-6 and an identity of 30% or less conserved domains were used as cut off. Furthermore, proteins were analysed with HHpred[62] (Hidden Markov Model Database: pdb 70_01Mar16 + SCOPe95_2.05; MSA generation Method: PSI-BLAST; with 3 iterations; http://toolkit.tuebingen.mpg.de/hhpred) for

structure homology analysis. Subcellular protein localization was predicted using PSORTb 3.0 (http://www.psort.org/psortb/)[63] with the Gram-negative option. Detected proteins were further compared to our bioinformatic analyses, as well as to previous analyses[23] (Supplementary Table 2).

**Construction of the *P. limnophila* ΔMC mutant.** We searched for homologues of the *G. obscuriglobus* MC protein gp4978 (WP_010038441.1) in the *P. limnophila* genome using BLAST. Plim_1972 (WP_013110233) was found to be most similar (E value 0, similarity 40%). In addition, the structure-based analysis for MC-like proteins was repeated for *P. limnophila*, as previously described[15], leading to the identification of other potential MC proteins in addition to Plim_1972 (Supplementary Fig. 9). Due to the highest sequence similarity, Plim_1972 (the only gp4978 homologue with an NCBI BLAST alignment score above 200) was selected for gene deletion.

For Red/ET Recombination (GeneBridge) based deletion of the *P. limnophila* MC homologue, a fosmid library was prepared using EpiFOS Fosmid Library Production Kit (Epicentre). Approximately 1,377 clones were screened for the presence of Plim_1927 (PLIM_RS10200). Four positive clones were found, and clone P12H10 was further processed with the BAC Subcloning kit (GeneBridge). As linear template, the kanamycin resistance cassette from pCJ003 was amplified (primer: CJ590: 5′-GCC GCC GTT TCT AAT TGA ACT ATG CCA TTC TGA TGA TCG AGA TTC AGT TCG CGG TTT TAT GGA CAG CAA G-3′ and CJ591: 5′-TTC ACA TGT GTT TTC TCG ATA ATG AGC ATT TTT TGA TGA GAA TCT GCG ATG CGA CAC GGA AAT GTT GAA T-3′). After electroporation, five clones were tested for carrying the selective marker and the Plim_1972 gene. The insertion and insertion sides were tested using primers CJ590, CJ591 and CJ600 (TTT GAA TGG CGA CTG ATC GG), CJ 601 (GGT TTC TTC CAT GAA GTC CAG TT) respectively. Three positive clones were identified (pCJ0100-pCJ0102). pCJ0100 plasmid DNA was linearized using *Hind*III (Fermentas). Linear DNA (2–4 μg) was introduced into *P. limnophila* using electroporation, as described before[25]. Cells were incubated on M3 plates (30 μg ml⁻¹ kanamycin & 30 μg ml⁻¹ chloramphenicol) for 7 days at 28 °C. Eight selected clones were incubated with M3 (30 μg ml⁻¹ kanamycin and 30 μg ml⁻¹ chloramphenicol) for 8 days at 28 °C at 90 r.p.m. Clone 6 (ΔPlim_1972) was used for further investigations and subsequently named *P. limnophila* ΔMC.

In order to validate the genetic background of *P. limnophila* DSM 3776 in comparison to *P. limnophila* ΔMC, two different PCRs (Taq polymerase Qiagen), targeting either the flank regions of gene Plim_1972 (primers: CJ787_F 5′-CGA AAC CGC TTG AAG ATG A-3′ and CJ788_R 5′-AAT ACA CAC CCA TGT GTT GTT GC-3′; 3,628 bp amplicon for WT and 1,295 bp for ΔMC) or the kanamycin cassette, that is unique to the ΔMC mutant (primer: CJ326a 5′-GCG GTT TTA TGG ACA GCA AG-3′ and CJ327a 5′-GCG ACA CGG AAA TGT TGA AT-3′; 1 kb amplicon) were performed (Supplementary Fig. 10a,b). To further validate *P. limnophila* ΔMC, an exponentially growing culture of WT and mutant cells was used to prepare protein extracts according to ref. 64. Samples were heated to 99 °C for 5 min and subsequently separated on a 12% gel SDS–PAGE at 80 V for 20 min, following by 100 V for 1.5 h. PVDF membrane (Carl Roth) was equilibrated for 10 s in methanol and washed in ddH₂O. SDS–PAGE gel and membrane were incubated for 10 min in transfer buffer and blotted in transfer buffer for 2.5 h at 90 V at 4 °C. The membrane was blocked with skim milk powder (0.1 g l⁻¹) over night at RT. Anti-Plim_1972 IgG (rabbit; COVANCE, Denver, USA) was incubated as first antibody (dilution 1:1,000) for 1 h at RT. The membrane was washed three times with PBST. The horseradish peroxidase conjugated second anti-rabbit-antibody (dilution 1:500; Cell Signalling Technologies) was incubated for 1 h at RT. Afterwards, the membrane was washed for three times with PBST. For detection, a horseradish peroxidase substrate system Lumi-light (Roche) was used. The blot was analysed using the photo imager Fujifilm LAS-3000.

**Macromolecule uptake experiment.** For GFP uptake, 1 ml of a stationary phase liquid culture of *G. obscuriglobus* and *P. limnophila* were harvested and incubated as previously described[16]. Cells were washed after incubation with 1 ml tap water and further analysed using WF fluorescence light microscopy and SR-SIM. As negative control, cells were poisoned at 50 μM CCCP final concentration. For co-localization, samples were stained afterwards with Nile Red and DAPI. To determine a positive uptake, treated cells were compared to the negative control.

To visualize the uptake of fluorescein-labelled dextran (40 kDa, Thermo Fisher) or Alexa 647 labelled dextran (10 kDa, Thermo Fisher), *P. limnophila* and *G. obscuriglobus* cells were incubated in growth medium for 1 h at 28 °C with a final concentration of 5 μg ml⁻¹ dextran. Cells were either washed and analysed directly after incubation or stained with Nile Red and DAPI.

For dSTORM experiments a 0.01% Polylysine coated glass bottom dish (MatTek) with Alexa 647 labelled dextran at a final concentration of 5 μg ml⁻¹ served as negative control. The SR-Tesseler method was used for cluster analysis in control and *P. limnophila/G. obscuriglobus* feeding experiments employing a minimum range of 40 nm and a minimum localization of 20 as previously described[65].

To compare the uptake of *P. limnophila* WT and the ΔMC mutant, 10 ml cells were inoculated in 20 ml M3 medium and incubated for 6 days at 28 °C with slight agitation (90 r.p.m.). These cultures were divided into triplicates, diluted (1:3) with fresh M3 medium and incubated for 6 days at 28 °C. Subsequently cells were fed

with 1 μg ml⁻¹ dextran (10 kDa) and incubated for 1.5 h. Unlabelled cells and cells treated with 50 μM CCCP served as control. Subsequently cells were stained with DAPI and analysed using fluorescence microscopy. A minimum of 175 cells in three different fields of view for each replicate were analysed (in total 15,556 WT and 19,502 mutant cells).

For SEM and TEM uptake experiments 1 ml of *P. limnophila* cells were inoculated 1:3 in M3 medium and incubated 4 days at 28 °C with slight agitation (90 r.p.m.) in a baffled flask. Cells were incubated with 100 μl of GP10-DX-20 (Gold Nanoparticles, Dextran Coated, 10 nm, 20 ml) in 1 ml 10 mM Tris pH 7.5 for up to 4 h at 28 °C. Subsequently cells were washed two times in sterile tap water and then fixed in 1% formaldehyde for 1 h at 4 °C.

**Direct stochastic optical reconstruction microscopy (dSTORM).** dSTORM reconstruction was acquired with a Nikon Eclipse Ti (Nikon GmbH) inverse microscope in 'Sedat' configuration, with 647 nm Fibre Laser and 405 nm Argon laser and a mercury vapour and bright field LED. Differential interference contrast (DIC) was used for WF overlays. Images where taken with the Nikon × 100/1.49 Oil APO TIRF objective and an ANDOR iXon3 camera. dSTORM-MEA-buffer was used as previously described[55]. A time series of 5,000–20,000 frames per image was recorded at ∼55 Hz with 300 mV excitation and was processed with NIS- Elements Imaging software 4.11 (Nikon). Minimum peak height was determined according to the signal intensity. Drift correction was applied for all analysed images.

**TEM and SEM analysis of planctomycetal appendices.** For negative staining thin carbon support films were prepared by sublimation of carbon onto a freshly cleaved mica surface. Samples were negatively stained with 0.5–2% (w/v) aqueous uranyl acetate, pH 5.0, air-dried and examined in a TEM 910 transmission electron microscope (Carl Zeiss AG) at an acceleration voltage of 80 kV. Images were taken at calibrated magnifications using a line replica. Images were recorded digitally with a Slow-Scan CCD-Camera (ProScan, 1,024 × 1,024) with ITEM-Software (Olympus Soft Imaging Solutions).

For the detection of gold labelled dextran, 4 μl drops of the bacterial solution were placed onto Butvar-coated 300 mesh copper grids, allowed to settle for 5 min and washed with distilled water. After air drying, samples were coated with a thin carbon layer and examined in a field emission scanning electron microscope (Zeiss Merlin, Carl Zeiss AG) using the Everhart Thornley HESE2-detector at an acceleration voltage of 5 kV and at calibrated magnifications.

**Data availability.** The mass spectrometry proteomics data have been deposited in the ProteomeXchange Consortium database via the PRIDE (ref. 66) partner repository with dataset identifier PXD005738. The authors declare that all other data supporting the findings of this study are available within the paper and its Supplementary Information Files, or from the corresponding author on request.

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

## Acknowledgements

We acknowledge Anja Heuer (DSMZ Braunschweig) and Bastian Kruse (TU Braunschweig IB23 class of 2015) for skilful technical assistance. We thank Robert Barnett for proof-reading the manuscript, Volker Müller (Goethe University Frankfurt) for providing the anti-Na$^+$-F$_1$F$_0$-ATPase antibody and Dimitry Tegunov (MPI Martinsried) for the development of the frame-alignment tool. This work was supported by the Deutsche Forschungsgemeinschaft (DFG grants JO 893/2-1 and JO 893/3-1) and the Max Planck Society. M.C.f.v.T. was supported by ERC AG 2,32,937.

## Author contributions

C.J. designed the study with help from C.B. and R.K. C.B., P.R. and D.B. cultivated all bacteria. M.J. constructed the GFP mutant. D.P.D. identified the MC protein homologues in *P. limnophila* and helped selecting the gene for deletion. C.J. constructed the *P. limnophila* ΔMC mutant, and C.B. and P.R. performed the verification. O.J. performed the bioinformatics analysis. C.B. did the staining- and dextran/GFP feeding experiments, the WF/dSTORM imaging and designed all figures together with C.J. and with help from all other authors. C.B. and D.B. performed the sucrose treatment and the immunofluorescence microscopy. C.B., G.R. and R.A. performed the SR-SIM and 3D reconstruction. Ma.S., Mi.S and H.E. performed the cryo-electron tomography including micromachining, segmentation and 3D reconstruction. M.R. accomplished the immunogold labelling and TEM/SEM imaging with help from C.B. and C.J. M.C.F.v.T. and L.v.N. performed the freeze-etching experiments and subsequent TEM analysis. C.B., M.K. and S.E. carried out the proteome analysis. C.J. and C.B. wrote the manuscript with help from all authors.

## Additional information

**Competing interests:** The authors declare no competing financial interests.

