## [Peer Review File · Nature Communications]

Reviewers' comments:

Reviewer #1 (Remarks to the Author):

There the authors use a variety of techniques to analyse the structure of Planctomycetes. While I can't really comment on the novelty of the work from a biological point of view, technically, the combination of techniques is nicely done. The problem I have with the paper is the lack of quantification throughout the manuscript. As it is, it is quite descriptive with phenomena described "by eye". This is probably addressable for the most part and, assuming their statements hold up after quantification and statistical analysis, I would say the manuscript is worthy of publication. While the whole manuscript needs quantitative analysis, p values on any stated changes etc, here are some examples:

Figure 1 for example: I find the discussion of the images too descriptive and this should be made more quantitative. For example, how were the cells sorted into the 3 different morphology groups - was this by eye or could some image analysis algorithm be used to make this more objective? What percentage of the cells fell into each of the three morphologies and what about borderline cases? There is probably some quantification that could be done of the image too. In the best case scenario this would be some kind of automated morphology analysis but at the very least, some line profiles across the cell might better illustrate the membrane/cytosolic staining patterns as well as to demonstrate the increase in resolution from SIM.

To give another specific example of quantification: SI Fig 2, the author mention the "shrunken nucleus" this could easily be quantified by nucleus area or diameter in ImageJ (for example) and the differences evaluated by p-values to check significance. In another example, membrane invaginations might be quantified by the percentage of membrane fluorescence overlapped with cytosolic fluorescence.

Another example is the description of the dSTORM imaging (Fig 5) where the authors report an "almost homogeneous distribution". This is quite a difficult statement for dSTORM. Firstly, again, this is not quantified and there are numerous "cluster analysis" tools the authors could use to quantify the degree of clustering/homogeneity. Ripley's K-function, Pair correlation analysis, Bayesian cluster analysis and Voronoi Tessellation are some of these. It should be remembered though that the "multi-blinking" phenomenon in dSTORM causes artificial clustering and so results need to be interpreted with caution. Good positive and negative clustering controls (dextran randomly coated on a plate?) might be useful if possible. Also, some more detail in the methods section is necessary rather than just "images were processed in NIS". What were the settings? Photon filters, drift correction, overlapped PSFs etc.

Finally, as a minor point: The authors state that SIM "doubles the resolution" of conventional imaging. This is really the best case scenario. It would be useful for the authors to measure the resolution, e.g. by imaging small fluorescent beads or by a line profile across a thin sample - e.g. membrane or actin fibre.

Reviewer #2 (Remarks to the Author):

The authors apply new genetic and microscopic tools to investigate the nature of the cell organization of members of the Planctomycetes. They find that these organisms exhibit characteristics normally associated with gram negative bacteria and conclude that the Planctomycetes are likely variants of this well known class of bacteria rather than being a group that represents an intermediate stage between prokaryotes and eukaryotes. The overall weight of the evidence seems to strongly support the authors' conclusions. The inner and outer membranes are more clearly visualized, supporting the "diderm" interpretation of cell organization. Also, the

uptake of large particles via fibrous structures instead of via phagocytosis seems convincing. The major deficiency in the presentation is adequate quantification of the observations they have amassed.

Major Comments

1. The authors have observed many hundreds of cell pictures to arrive at their conclusions. However, in each instance where they report results of this sort, they include only one or two "example" pictures to illustrate what they say are the typical results for each class of result. The paper would be much more convincing if the authors were to report the results in a more quantitative way. With an increasing reliance on visual interpretation of microscopic evidence has come a lamentable lack of quantification, leading to potentially misleading conclusions. The authors need to address this throughout. The following are some examples of this practice, not a comprehensive list.

a) p 2 lines 57-60 ... "Thousands of cells" were analyzed, leading to the three morphotypes the authors describe. These results should be quantified in some way, and that method should be described clearly and perhaps illustrated. The distribution of cells among these morphotypes should be reported, along with any results that do not fit these post-facto categories.

b) p 2 lines 84-87 ... The invaginations of the inner membrane are said to "increase significantly". How many times did this occur? What is the meaning of "significant"? What are the before and after measurements? Without these quantitative measurements across a number of observations, there is no way for the reader to evaluate the strength of these statements or the conclusions derived from them.

c) p 3 lines 52-55 ... The "vast majority" of co-localization results implies that the results can be quantified. They should be quantified, the method of quantification should be clearly explained, and the results should be reported.

d) p 5 lines 13-14 ... Again, what does the "vast majority" mean? How was this quantified? What data supports this kind of assertion? What are the statistics?

e) p 5 Figure 3II ... The absence of statistical treatment is especially evident here.

f) p 6 lines 6-8 ... Only a few cells took up GRP out of the many (30?) that can be viewed in this single view. The data should be quantified, and the results with and without CCCP should be compared.

g) p 6 lines 81-87 ... 110 micrographs were examined - and dextran molecules were "primarily" associated with fibres (lines 82-82), and this association was "much less" when the cells were treated with the energy-depleting CCCP (lines 84-86). Once again, the reader cannot evaluate the strength of these assertions without explicit quantification in a transparent way. In addition, the pictures of *G. obscuriglobus* show few if any obvious gold particles, so there is no way for the reader to determine if there is a difference between the two conditions.

h) p 7 Figure 4 legend, line 3 ... How many cells were observed to find that the dextran signals were "exclusively detected in -CCCP cells"?

2. There are two separate problems associated with the results as reported and interpreted regarding the distribution of the "vast majority of fluorophores associated with the innermost membrane" (p 5 lines 13 and following, and Figure 3).

a) First, the fact that an immunogold particle can be seen to touch one of the membranes is not, in and of itself, proof that the particle is actually in contact with that membrane. It may be that the

particle is above or below. The authors should explain why this situation may or may not occur. Are the preparation and labeling techniques sufficiently accurate to rule this out?

b) Second, and most importantly, the authors report in the legend to Fig. 3 that they observed 400 counts associated with the innermost membrane, 73 counts in the outer membrane, and 93 counts elsewhere. Thus, they find that the counts in the "innermost membrane" exceed counts in the "outer membrane" by a ratio of about 5:1. However, if each cell contained 5 times as much innermost membrane as there was outer membrane, then this is exactly the ratio that would be expected if the distribution of particles was simply random. That is, if the particles were staining ATPase molecules in ANY membrane, then more would naturally end up in innermost membranes if the volume of those membranes was dominate (which it seems to be). The authors should quantify the volume (or, probably, the "lengths") of each type of membrane in their pictures, and only then re-evaluate whether or not the distribution represents a random distribution between membrane types. Simply stating that a "vast majority" are localized one way or another is not accurate in this situation.

3. In Figure 1, it is not clear that the B and C models are really different. Might they simply be different views of one type, with some invaginations appearing wrapped around the cytoplasm whereas other invaginations are not?

4. The authors find ten putative membrane coat-like particles in their organism (p 6 lines 38-40, and lines 46-49). They delete only one of these as a test to determine if it might be responsible for dextran internalization, and they conclude that this Planctomycete does not use MC proteins for this purpose (p 6 lines 65-67). This conclusion is unwarranted. The most that can be said is that this particular MC homologue is not involved in dextran internalization. Even if this particular homologue is the most likely candidate, it is impossible, using bioinformatic information alone, to be certain that other MC candidates might not be involved. The sweeping conclusion should be removed.

5. p 8 lines 71-73 ... The proposal that the crateriform structures are the sites of dextran internalization is highly speculative and is supported by no data at all. Even if just "proposed", the authors should refrain from making this claim just yet.

6. Supplemental p 3, Figure S2 (I-C) ... Where are the cells that have NOT been osmotically stressed (so that the size of the periplasmic bays can be compared)?

Minor comments

7. Can the authors distinguish dividing cells from those that are not dividing? If so, does the structure of these cells correspond with that expected of gram negative bacteria?

8. In the text, please include the "I" and "II" or "A" and "B" (etc) designations when referring to data in the figures (in all cases). The reader should not have to figure out where to look for the specific information.

9. Figure 1 ... In the legend, please explain the "SR-SIM" abbreviation. This is the first use of this term in the figures.

10. Figure 2 ... The PG is said to be represented by an orange line. This color and the line itself are not visible.

11. Unless the authors performed biochemical tests, then they cannot say that these cells had a "confirmed ... peptidoglycan layer" (p 7 lines 11-13). They might say they observed something that was "consistent" with such a layer, but if so, then the reason for drawing this conclusion should be stated clearly.

12. The DIC images in Figure 4 (IB) are confusing. What information is this supposed to convey?
13. p 9, Figure 5 (I) ... Except for the center figure (B), it is impossible to see the structures that are supposedly being singled out. Also, are the structures in II(J) really those seen in these photos? There is no way to tell.
14. Figure S3 ... The legend refers to A and B parts, but the figure is labeled I and II.
15. Figure S5 ... Do the colors in the top part (the secretory drawings) mean anything? Are they supposed to be connected with the colors in the lower part? Also, the colors in the lower section are difficult to see, and those that can be seen do not help the reader interpret the distribution of these proteins. I believe that listing these in a more classical Table format would clarify which proteins were or were not present.
16. Figure S6 (III) ... The green GFP is said to stain the cytosol, but then later the GFP signal is said to represent the periplasmic space. Which is it? This is definitely confusing.
17. Figure S7 (I-B) ... The green signal is invisible or too small to be of use.
18. Figure S8 ... The "Plim" designations are too small to be seen easily. Why not refer to the proteins by the large numbers that are more easily read (e.g., "D5SXW5" in this case)?

Reviewer #3 (Remarks to the Author):

The manuscript "Determining the cell biology of Planctomycetes" by Boedeker et al is an interesting analysis of three aspects of Planctomycetes physiology: the body plan of the envelope, the localization of ATPases, and uptake of various molecules. The analysis relies heavily on advanced imaging (SIM, STORM, and EM), and is complemented by genetics in one of the organisms. I am not an expert on the ongoing controversies regarding eukaryogenesis and how this related to the Planctomycetes, though I think their discussion of this topic is quite measured, focusing instead on the conclusions motivated by their data. The imaging is quite impressive, and some of the cells portray really beautiful internal organization.

In general, I found the manuscript to be clear, well written, and very thorough through the application of multiple imaging modalities. The topic is certainly of broad interest, and a systematic study of this group of organisms is important for opening the door to further biological studies, and I'm mostly convinced that endocytosis is not the mechanism of uptake, which is the major finding of the paper. However, there are a few major questions I have about the conclusions:

- My major concern has to do with the bioinformatic argument about Plim_1972, the putative MC protein. They say that this is the homologue since it has 40% similarity vs 22-29% for the other 9, but this argument would fall apart in many cases. For instance: MreB has much less than 16% homology to actin, yet is a structural homolog with many similar properties. Perhaps more applicable: there are 3 MreB homologs in *B. subtilis*, and it is still not clear which is the one(s) that behave similarly to MreB in *E. coli*. Furthermore, there are many instances in which the assumed enzymatic activity based on homology turns out to be wrong. Basically, 40% vs. 22% is not enough to conclude that they've got the right MC, leaving open a large door of 9 other putative proteins.

- Also, it's worth noting that the statement that dextran binds to fibres that originate from the large crateriform structures does not exclude endocytosis, though I agree that their cryoEM argues against endocytosis.

- In Fig. 1, I think I may be missing what they want me to notice. I understand the argument

about the enlarged periplasmic space, but what is it they want me to notice in the phase contrast images (and why do they appear blue in my PDF? It makes it seem as if it's a DAPI overlay, but that can't be true). How am I supposed to see that the structures are connected to the OM?

- Similarly, I just don't see from the SR-SIM how I'm supposed to see that the membrane structures are periplasmic spaces - looks like GFP is actually inside these spaces in some cases (e.g., in B)

- For the proteome measurements, seems odd that there are so few OM proteins (<1%) - what is the total abundance of OM proteins out of their spectral counts? I'm worried that they're missing a lot of material and hence the numbers are not so meaningful.

- I'm confused about the appendages (I don't think "appendices" is the word that they're meaning to use): I would think that if they are membrane-bound, then they would appear in light microscopy images. This brings up a larger point, that the recent history of papers about appendages (e.g. from Sigal Ben-Yehuda's lab on "nanotubes") have not been well received, with many suggesting that they are TEM artifacts. Would be good to have separate confirmation from light microscopy, which seems like it would be obvious based on their size.

- As far as I can tell, the authors do not give info about the growth rates, nor which OD the cells are being imaged at in the Methods. This would be very useful knowledge.

Please note that I'm not suggesting that they need to go and knockout the other 9 genes to address the MC protein question. I think in the sum total of all of their experiments, they can certainly argue convincingly against endocytosis. However, it would be nice to have a richer analysis of these genes since I don't think this is going to convince people who work on MC proteins that there is no endocytosis in Planctomycetes.

Some minor comments and questions:

- have the authors tried correlated SEM/SIM? This seems like it would help to address question above about the appendages

- I'd like to see some discussion of the nucleoid compaction - are there similarities to other, more well studied organisms?

- In the introduction, there is a citation to "Rast et al, in prep" regarding a statement about the Planctomycetes having a proteinaceous cell wall; I assume the reference is a paper that shows that in fact the Planctomycetes have a PG cell wall. If so, I don't think this statement belongs here, there is no evidence in the paper to support this conclusion and the previous paper has not been even submitted. If not, this citation doesn't belong where it is, where it is modifying bacteria having PG cell walls.

- Would be useful to have a clearer description of the craters and what the authors are concluding: are they saying there are two types, and do these types differ as to whether the membrane is attached (and how do they determine membrane attachment?)

Reviewer #4 (Remarks to the Author):

Dear Authors of the manuscript NCOMMS-16-13455,

by a Senior Editor of Nature Communications I was asked to comment on the cryo-electron microscopy/tomography analyses included in your manuscript. Other parts of your research concerning planctomycetes biology would be reviewed by other referees; because the manuscript is under review for already some time I was asked to do it within 7 days (unfortunately, the Email is confidential - I am not allowed to cite it here as I originally did).

I agreed and got the manuscript 09/08/2016 10:42.

To keep the deadline (almost), I concentrated on the cryoEM part and can say: it fulfills all criteria of the world-renown top-level "Good (Baumeister) cryoEM-Laboratory Practice". More literature should be cited in "Materials and Methods" (as indicated in the annotated merged manuscript-file

98984_0_merged_pdf_1790422_m8gmzn_CH.pdf, that I send separately to the Editor - I thought that makes your life easier, and can/should be included in the new "transparent peer review system for new manuscripts submitted from January 2016" that I very much welcome/appreciate). I am now waiting for the next high-class paper from you showing subtomogram averaging data (from cryoET reconstructions of cryoFIB-SEM lamellae) of the "enigmatic" crateriform structures acting as buttons to connect inner and outer membrane (~100 per cell - it should be possible).

Here my comments along the Editor's criteria for reviewing:

1) What are the major claims of the paper?

They are all mentioned in the Abstract/Conclusions of the manuscript. (I annotated some proposals in the .pdf to strengthen it even further.)

2) Are they novel and will they be of interest to others in the community and the wider field?

They are revolutionary as they bring back Planctomycetes to their gram-negative bacterial roots (once again and even stronger). The study is not only important for microbiologists but also for everybody thinking/working in structural (cell) biology: it again shows that in particular emerging imaging technologies like cryoEM (here of thinned lamellae produced by cryoFIB-SEM) or superresolution (here 3DSIM) keeping the biological material in its best preservation state (vitreous for cryoEM) can help to correct wrong concepts. And that in an integrative manner building on newly developed genetic tools, additional (conventional) imaging techniques, proteomics and many more.

3) Is the work convincing, and if not, what further evidence would be required to strengthen the conclusions?

The paper already provides a wealth of convincing evidence (and appropriate statistics), nothing more is needed for the conclusions drawn.

4) On a more subjective note, do you feel that the paper will influence thinking in the field?

See point 2.

I like your manuscript very much, indeed. My recommendation to the Editor is: (very) minor revision and acceptance/publication asap.

I do not send any "Confidential remarks for the editor only".

To the Editor: Thanks for giving me the opportunity.

Kind regards

Christoph Hagen

(Oxford, 17/08/2016, ~1am)

PS: Due to the Editor's request to leave out confidential material (Email 18/08/2016, 09:26) and my visit of the extremely interesting XRM2016 today here in Oxford I was only now (18/08/2016, ~20:30) available to edit and resend the review. Sorry for the delay.

‘Reviewer #1 (Remarks to the Author):

There the authors use a variety of techniques to analyse the structure of Planctomycetes. While I can't really comment on the novelty of the work from a biological point of view, technically, the combination of techniques is nicely done. The problem I have with the paper is the lack of quantification throughout the manuscript. As it is, it is quite descriptive with phenomena described "by eye". This is probably addressable for the most part and, assuming their statements hold up after quantification and statistical analysis, I would say the manuscript is worthy of publication.’

We agree with Reviewer #1 who's comment is in line with Reviewer #2. Thus, we reworked the entire manuscript and added as much quantification as possible.

However, on an organismic level in some cases objective quantification is difficult to obtain and “by eye” analysis is the only option. Please consider that we wanted to proof endocytosis and compartmentalization in Planctomycetes (application to the DFG for funding of this project available on request). During multiple control experiments, we first realized the presence of peptidoglycan and now the absence of endocytosis and compartmentalization. Thus, we followed good scientific practice in trying to falsify our hypothesis which unexpectedly worked out. Thus, we are sure that representative images were selected.

However, to gain maximal objectivity in such cases multiple investigators analyzed the data independently and details of this analysis is provided as supporting information (e.g. ATPase immunogold localization experiments).

‘While the whole manuscript needs quantitative analysis, p values on any stated changes etc, here are some examples: Figure 1 for example: I find the discussion of the images too descriptive and this should be made more quantitative. For example, how were the cells sorted into the 3 different morphology groups - was this by eye or could some image analysis algorithm be used to make this more objective? What percentage of the cells fell into each of the three morphologies and what about borderline cases? There is probably some quantification that could be done of the image too. In the best case scenario this would be some kind of automated morphology analysis but at the very least, some line profiles across the cell might better illustrate the membrane/cytosolic staining patterns as well as to demonstrate the increase in resolution from SIM.’

As automated analysis did not work for our images we decided to distinguish only two different morphotypes, with- or without invaginations, acknowledging that such invaginations can differ. This yes/no decision was easy to take and we used the suggested line profiles to visualize this differences (Fig. 2 II). We thank Reviewer #1 for this great suggestion. In addition, we provide an example overview image (Fig. 2 I) along with detailed information on the different morphotypes in each field of view (Fig. S1 I). Detailed numbers are provided in the modified manuscript as well. For example, 1838 cells were analyzed although we display only three representative cells for the sake of clarity

Furthermore, we used this technique to demonstrate the increase in resolution during SIM experiments comparing the measured membrane thickness (Fig. S2 II a+b). These results were further verified employing DNA origamis with fluorophores in a distinct distance of 120 nm (Fig. S2 II c+d). Thus, the real resolution of this particular microscope is slightly above 120 nm as fluorophores in 120 nm distance were not fully resolved. For the sake of clarity, we reworded the manuscript ‘... roughly doubles the resolution of WF...’ as we provide details in the supporting information section.

‘To give another specific example of quantification: SI Fig 2, the author mention the "shrunken nucleus" this could easily be quantified by nucleus area or diameter in ImageJ (for example) and the differences evaluated by p-values to check significance.’

We modified figure and manuscript accordingly and provide the requested p-values.

‘In another example, membrane invaginations might be quantified by the percentage of membrane fluorescence overlapped with cytosolic fluorescence.’

In general, such a quantification is technical possible (for example, we applied such a method to quantify ATPase localization in WF in the revised version of our manuscript). However, since we do not distinguish between three-, but only two morphotypes now (with- and without invaginations) we simply quantified the amount of both morphotypes instead. As our CET data shows, there are huge variations within a planctomycetal population ranging from cells that display no- to cells that display multiple invaginations. Thus, to determine overlaps among multiple cells will not provide meaningful results. In a follow-up project, we will try to understand what triggers invaginations. For this studies Reviewer #1’s suggestion is very valuable.

‘Another example is the description of the dSTORM imaging (Fig 5) where the authors report an "almost homogeneous distribution". This is quite a difficult statement for dSTORM. Firstly, again, this is not quantified and there are numerous "cluster analysis" tools the authors could use to quantify the degree of clustering/homogeneity. Ripley's K-function, Pair correlation analysis, Bayesian cluster analysis and Voronoi Tessellation are some of these. It should be remembered though that the "multi-blinking" phenomenon in dSTORM causes artificial clustering and so results need to be interpreted with caution. Good positive and negative clustering controls (dextran randomly coated on a plate?) might be useful if possible. Also, some more detail in the methods section is necessary rather than just "images were processed in NIS". What were the settings? Photon filters, drift correction, overlapped PSFs etc.’

We entirely agree with Reviewer #1 and modified the manuscript accordingly. The three pages long additional supporting Figure 11 was added that summarizes control experiments. For the sake of focus we only slightly modified Figure 5. The controls include both, a negative control with fluorescent dextran plus a cluster analysis for the negative control of both *P. limnophila* and *G. obscuriglobus* feeding experiments. The negative control did not result in any random clustering. The feeding experiments show intense clustering, but not within the range important for the predicted vesicles (50–200 nm). Thus, we can even more convincingly conclude that there are no vesicles. A finding nicely supported by our CET data.

The manuscript was revised accordingly and details of the analysis are provided in the material and method section.

‘Finally, as a minor point: The authors state that SIM "doubles the resolution" of conventional imaging. This is really the best case scenario. It would be useful for the authors to measure the resolution, e.g. by imaging small fluorescent beads or by a line profile across a thin sample - e.g. membrane or actin fibre.’

We agree with Reviewer #1 and measured the resolution with DNA origamis and in direct comparison of membrane signals (fluorescence intensity line profiles) as mentioned above.

‘Reviewer #2 (Remarks to the Author):

The authors apply new genetic and microscopic tools to investigate the nature of the cell organization of members of the Planctomycetes. They find that these organisms exhibit characteristics normally associated with gram negative bacteria and conclude that the Planctomycetes are likely variants of this well known class of bacteria rather than being a group that represents an intermediate stage between prokaryotes and eukaryotes. The overall weight of the evidence seems to strongly support the authors' conclusions. The inner and outer membranes are more clearly visualized, supporting the "diderm" interpretation of cell organization. Also, the uptake of large particles via fibrous structures instead of via phagocytosis seems convincing. The major deficiency in the presentation is adequate quantification of the observations they have amassed.’

We highly appreciate Reviewer #2’s valuation of our manuscript. The criticism is in line with the suggestions of Reviewer #1 and we revised the entire manuscript to add as much quantification as possible and / or biologically relevant.

‘Major Comments

1. The authors have observed many hundreds of cell pictures to arrive at their conclusions. However, in each instance where they report results of this sort, they include only one or two "example" pictures to illustrate what they say are the typical results for each class of result. The paper would be much more convincing if the authors were to report the results in a more quantitative way. With an increasing reliance on visual interpretation of microscopic evidence has come a lamentable lack of quantification, leading to potentially misleading conclusions. The authors need to address this throughout. The following are some examples of this practice, not a comprehensive list.’

We agree with Reviewer #2 and revised the entire manuscript and the figures accordingly. For the sake of focus not all primary data can be shown, but we included overviews, more examples and enhanced quantification throughout the manuscript.

‘a) p 2 lines 57-60 ... "Thousands of cells" were analyzed, leading to the three morphotypes the authors describe. These results should be quantified in some way, and that method should be described clearly and perhaps illustrated. The distribution of cells among these morphotypes should be reported, along with any results that do not fit these post-facto categories.’

As explained above for Reviewer #1 we distinguish now only between two morphotypes with- and without invaginations. Furthermore, we modified Figure 2, now providing an overview and fluorescence intensity line profiles for different color channels. In addition, numbers are provided and counting results are given to ensure quantification (Fig. S1 Ia+b).

‘b) p 2 lines 84-87 ... The invaginations of the inner membrane are said to "increase significantly". How many times did this occur? What is the meaning of "significant"? What are the before and after measurements? Without these quantitative measurements across a number of observations, there is no way for the reader to evaluate the strength of these statements or the conclusions derived from them.’

We added several panels to Figure S1 (III +IV) and provide precise p-values and numbers in the manuscript. The material and method section explains in detail how measurements and calculations were performed.

‘c) p 3 lines 52-55 ... The "vast majority" of co-localization results implies that the results can be quantified. They should be quantified, the method of quantification should be clearly explained, and the results should be reported.’

We analyzed co-localization of DiOC₆(3) and FM4-64 signals by plotting the corresponding intensities against each other. The resulting graphs are shown in Figure S4 II and demonstrate a correlation. This was further verified by calculating Pearson’s correlations and Mander’s overlap’s for all analyzed species. The entire paragraph in the manuscript was revised accordingly and details of the methods are given in the material and method section.

However, from a biological perspective any green signal from the inside of the cell proofs that the innermost membrane is energized. Thus, we are very confident with our results that are in line with ATPase localization and CET analysis.

'd) p 5 lines 13-14 ... Again, what does the "vast majority" mean? How was this quantified? What data supports this kind of assertion? What are the statistics?'

We agree with Reviewer #2 and revised the entire Figure 4 (previously Fig. 3) and provided numbers and statistics. Furthermore, details of quantification are given in Fig. S5. The entire paragraph of the manuscript was rewritten.

'e) p 5 Figure 3II ... The absence of statistical treatment is especially evident here.'

We provide detailed statistics in Figure 4 I and employ dSTORM here as an additional proof to visualize our finding of Figure 4I. Thus, we are not sure if an additional statistical analysis is required given that we provided such data in Fig. 4 I and II plus details in Figure S5 and are 100% confident with our findings that are in line with CET data as well. However, if the reviewer insists we will provide more statistics.

'f) p 6 lines 6-8 ... Only a few cells took up GRP out of the many (30?) that can be viewed in this single view. The data should be quantified, and the results with and without CCCP should be compared.'

We agree with Reviewer #2 and modified the manuscript as follows:

*"We therefore revisited previously performed GFP uptake experiments with *G. obscuriglobus*²⁸. In 10.15% of the 1281 analysed cells we observed GFP uptake and the formation of GFP-containing foci in WF fluorescence microscopic experiments. Treatment of the cells with CCCP prevented uptake (0% uptake among 1294 analysed cells), indicating an active process (Fig. S7 II) as previously described²⁸."*

*'g) p 6 lines 81-87 ... 110 micrographs were examined - and dextran molecules were "primarily" associated with fibres (lines 82-82), and this association was "much less" when the cells were treated with the energy-depleting CCCP (lines 84-86). Once again, the reader cannot evaluate the strength of these assertions without explicit quantification in a transparent way. In addition, the pictures of *G. obscuriglobus* show few if any obvious gold particles, so there is no way for the reader to determine if there is a difference between the two conditions.'*

We entirely agree with the Reviewer and tried very hard to perform additional experiments to further strengthen the energy dependent binding. However, we received quite conflicting results in multiple independent experiments. Since many comments from all Reviewers addressed our Figure 6, we entirely modified it and do not mention the energy dependent binding of dextran to the fibers anymore. However, we are still 100% convinced that dextran binds to the fibers. We will address this issue in the future in detail.

'h) p 7 Figure 4 legend, line 3 ... How many cells were observed to find that the dextran signals were "exclusively detected in -CCCP cells"?''

We agree with the Reviewer and added the following text to the Figure legend:

“While representative images are shown, in total 1065 cells (7 fields of view) were analysed and 23% of them showed dextran uptake. In contrast, among 816 CCCP treated cells not a single uptake event was observed. Bar indicates 1 μm .”

‘2. There are two separate problems associated with the results as reported and interpreted regarding the distribution of the "vast majority of fluorophores associated with the innermost membrane" (p 5 lines 13 and following, and Figure 3).

a) First, the fact that an immunogold particle can be seen to touch one of the membranes is not, in and of itself, proof that the particle is actually in contact with that membrane. It may be that the particle is above or below. The authors should explain why this situation may or may not occur. Are the preparation and labeling techniques sufficiently accurate to rule this out?

‘b) Second, and most importantly, the authors report in the legend to Fig. 3 that they observed 400 counts associated with the innermost membrane, 73 counts in the outer membrane, and 93 counts elsewhere. Thus, they find that the counts in the "innermost membrane" exceed counts in the "outer membrane" by a ratio of about 5:1. However, if each cell contained 5 times as much innermost membrane as there was outer membrane, then this is exactly the ratio that would be expected if the distribution of particles was simply random. That is, if the particles were staining ATPase molecules in ANY membrane, then more would naturally end up in innermost membranes if the volume of those membranes was dominate (which it seems to be). The authors should quantify the volume (or, probably, the "lengths") of each type of membrane in their pictures, and only then re-evaluate whether or not the distribution represents a random distribution between membrane types. Simply stating that a "vast majority" are localized one way or another is not accurate in this situation.’

We repeated all experiments and revised Fig. 4 and Fig. S5 to address Reviewer #2’s concerns. In the revised version of Fig. 4 we display localization uncertainty for each single gold particle as a circle. Color coding explains to which localization the particle was assigned. Results of the new experiments were analyzed by three researchers independently and the results of each individual analysis is given in Fig. S5 III. Each scientist counted at least 968 gold particles and we provide an overview in Figure S5 and Table S1. We assigned particles to either the outer-, inner membrane or to unspecific localization outside or inside of the cell. All three localizations other than at the inner membrane were statistically compared against the fraction that localizes at the innermost membrane and differences were found to be significant. P-values are given in Fig. S5.

Our analysis is comparable to what has been done before to localize ATPases in Planctomycetes and is in line with fluorescence localization employing the same antibody. The localization is significant in fluorescence experiments as well. Furthermore, DiOC₆(3) and FM4-64 studies revealed that the innermost membrane is energized. This was observed in other independent studies as well, but not acknowledged (see discussion). Thus, we are very confident that most, if not all ATPases localize at the innermost membrane.

‘3. In Figure 1, it is not clear that the B and C models are really different. Might they simply be different views of one type, with some invaginations appearing wrapped around the cytoplasm whereas other invaginations are not?’

We agree with Reviewer #2 and now distinguish only between two morphotypes (with- and without invaginations) and quantified them as well (now Fig. 2). CET data revealed that there are multiple variations

from non- to very complex invaginations and this will be part of a follow-up study.

‘4. The authors find ten putative membrane coat-like particles in their organism (p 6 lines 38-40, and lines 46-49). They delete only one of these as a test to determine if it might be responsible for dextran internalization, and they conclude that this Planctomycete does not use MC proteins for this purpose (p 6 lines 65-67). This conclusion is unwarranted. The most that can be said is that this particular MC homologue is not involved in dextran internalization. Even if this particular homologue is the most likely candidate, it is impossible, using bioinformatic information alone, to be certain that other MC candidates might not be involved. The sweeping conclusion should be removed.’

Although we are very confident about Plim_1972 being the homolog of the Gemmata MC that was involved in endocytosis-like uptake, we agree with the Reviewer that our conclusion went too far and modified the manuscript accordingly:

Results:

“Thus, our results demonstrate that *P. limnophila* does not employ the MC protein **Plim_1972** for dextran internalization.”

Discussion:

“**However, since *P. limnophila* comprises more than one MC-like gene, we cannot exclude that our mutant does not affect the proposed function in vesicle formation. Taken together,** we conclude that Planctomycetes do not use a **vesicle mediated** mechanism for macromolecule uptake into the periplasm.”

‘5. p 8 lines 71-73 ... The proposal that the crateriform structures are the sites of dextran internalization is highly speculative and is supported by no data at all. Even if just "proposed", the authors should refrain from making this claim just yet.’

We agree with the reviewer that our hypothesis about the crateriform structures is speculative. However, this is what proposals are, right? We do not agree that there is no evidence: How shall a macromolecule cross the outer membrane if porins are too small? We discussed this in detail in the manuscript. We agree that the proposal is mainly motivated by such theoretical assumptions, which we find convincing enough to undertake a complex follow-up project. This project was even suggested by Reviewer #4. Again, we propose this interpretation, we do not claim it. However, soon we will know more, but this will be another manuscript.

‘6. Supplemental p 3, Figure S2 (I-C) ... Where are the cells that have NOT been osmotically stressed (so that the size of the periplasmic bays can be compared)?’

We revised the entire Figure S1 (Fig. S2 before revision) and provide an overview of untreated and treated cells along with quantification and significant analysis. P-values are provided and the entire paragraph in the manuscript was revised.

‘Minor comments

7. Can the authors distinguish dividing cells from those that are not dividing? If so, does the structure of these cells correspond with that expected of gram negative bacteria?’

We can distinguish dividing- from non-dividing cells. Planctomycetes divide through polar budding lacking the otherwise universal major cell division protein FtsZ. In another study, we analyze their cell division that seems to differ from all 'normal' bacteria.

'8. In the text, please include the "I" and "II" or "A" and "B" (etc) designations when referring to data in the figures (in all cases). The reader should not have to figure out where to look for the specific information.'

We revised the entire manuscript accordingly.

'9. Figure 1 ... In the legend, please explain the "SR-SIM" abbreviation. This is the first use of this term in the figures.'

We revised the Figure legend accordingly.

'10. Figure 2 ... The PG is said to be represented by an orange line. This color and the line itself are not visible.'

We revised the Figure and provide a better image.

'11. Unless the authors performed biochemical tests, then they cannot say that these cells had a "confirmed ... peptidoglycan layer" (p 7 lines 11-13). They might say they observed something that was "consistent" with such a layer, but if so, then the reason for drawing this conclusion should be stated clearly.'

We did such experiments before (Jeske *et al.* Nature Communications 2015) and added the reference in the manuscript.

'12. The DIC images in Figure 4 (IB) are confusing. What information is this supposed to convey?'

The DIC images in the overlay proofs that the fluorescent signal localizes within the cells.

'13. p 9, Figure 5 (I) ... Except for the center figure (B), it is impossible to see the structures that are supposedly being singled out. Also, are the structures in II(J) really those seen in these photos? There is no way to tell.'

We agree with the Reviewer. As many suggestions of all Reviewers concerned Figure 6 (previously Fig. 5) we modified it and simplified the statement: There are two kinds of crateriform structures that give rise to two kinds of fibers. One is a stalk and attaches the cell to any given surface (shown for *P. limnophila* since *Gemmata* has no such stalk). The other type binds dextran and might serve as a molecular fishing rod (shown for both organisms). Additional requested light microscopic images proof along with our CET data that such

fibers are no artefact of TEM or SEM specimen preparation.

'14. Figure S3 ... The legend refers to A and B parts, but the figure is labeled I and II.'

We revised the Figure legend accordingly.

'15. Figure S5 ... Do the colors in the top part (the secretory drawings) mean anything? Are they supposed to be connected with the colors in the lower part? Also, the colors in the lower section are difficult to see, and those that can be seen do not help the reader interpret the distribution of these proteins. I believe that listing these in a more classical Table format would clarify which proteins were or were not present.'

We agree with the Reviewer that our color code was misleading. We revised the entire figure and provide two legends now for the different color codes in I and II. Furthermore, we redraw part II and color code and resolution were increased for better clarity.

'16. Figure S6 (III) ... The green GFP is said to stain the cytosol, but then later the GFP signal is said to represent the periplasmic space. Which is it? This is definitely confusing.'

These are two different experiments. First, a cytosolic GFP expressing mutant was generated. Accordingly, GFP localizes in the cytosol.

Second, GFP feeding experiments were performed. This GFP ended up in the periplasmic space. We modified wording to make this point clearer.

'17. Figure S7 (I-B) ... The green signal is invisible or too small to be of use.'

In Figure S7 uptake of GFP into the periplasm of *G. obscuriglobus* is shown. This is one example as 1281 cells in total were analyzed. We added quantification and error bars as well as a detailed description of the counting procedure. This figure is important as it proves GFP uptake in *Gemmata* that, under diffraction limiting conditions, looks like green foci, which fits to the vesicle interpretation. Only in super resolution the vesicles vanish.

'18. Figure S8 ... The "Plim" designations are too small to be seen easily. Why not refer to the proteins by the large numbers that are more easily read (e.g., "D5SXW5" in this case)?'

We agree with the Reviewer and redraw the entire Figure (now S9) to better resolution and readability.

‘Reviewer #3 (Remarks to the Author):

The manuscript "Determining the cell biology of Planctomycetes" by Boedeker et al is an interesting analysis of three aspects of Planctomycetes physiology: the body plan of the envelope, the localization of ATPases, and uptake of various molecules. The analysis relies heavily on advanced imaging (SIM, STORM, and EM), and is complemented by genetics in one of the organisms. I am not an expert on the ongoing controversies regarding eukaryogenesis and how this related to the Planctomycetes, though I think their discussion of this topic is quite measured, focusing instead on the conclusions motivated by their data. The imaging is quite impressive, and some of the cells portray really beautiful internal organization.

In general, I found the manuscript to be clear, well written, and very thorough through the application of multiple imaging modalities. The topic is certainly of broad interest, and a systematic study of this group of organisms is important for opening the door to further biological studies, and I'm mostly convinced that endocytosis is not the mechanism of uptake, which is the major finding of the paper. However, there are a few major questions I have about the conclusions:’

We highly appreciate the overall assessment of Reviewer #3.

‘- My major concern has to do with the bioinformatic argument about Plim_1972, the putative MC protein. They say that this is the homologue since it has 40% similarity vs 22-29% for the other 9, but this argument would fall apart in many cases. For instance: MreB has much less than 16% homology to actin, yet is a structural homolog with many similar properties. Perhaps more applicable: there are 3 MreB homologs in B. subtilis, and it is still not clear which is the one(s) that behave similarly to MreB in E. coli. Furthermore, there are many instances in which the assumed enzymatic activity based on homology turns out to be wrong. Basically, 40% vs. 22% is not enough to conclude that they've got the right MC, leaving open a large door of 9 other putative proteins.’

Although we in general acknowledge the difficulty of identifying homologue structure proteins, from our perspective we are convinced that Plim_1972 is the homologue to the *G. obscuriglobus* MC-like protein that was speculated to be involved in endocytosis-like uptake.

MreB is of bacterial-, actin is of eukaryotic origin. Thus, the phylogenetic distance is huge and since structure proteins are more conserved in structure than sequence, the low homology (16%) is not surprising. Basically, the same applies for MreB in *E. coli* and *B. subtilis*. Both organisms belong to different bacterial phyla and their phylogenetic distance is huge. Although we acknowledge that the phylum Planctomycetes within itself covers a huge phylogenetic distance, the organisms *P. limnophila* and *G. obscuriglobus* are by far closer related than *E. coli* and *B. subtilis*. Furthermore, we did not rely on sequence comparison alone, but predicted the structure as well (Fig. S10). For clarification, we reworked Fig. S10 and its Figure legend. The analysis was performed by the same scientist who identified the MC-like proteins in *Gemmata*.

However, to address Reviewer #3’s concerns, we revised the manuscript as follows:

Results:

“Thus, our results demonstrate that *P. limnophila* does not employ the MC protein **Plim_1972** for dextran internalization.”

Discussion:

“However, since *P. limnophila* comprises more than one MC-like gene, we cannot exclude that our mutant does not affect the proposed function in vesicle formation. Taken together, we conclude that Planctomycetes do not use a vesicle mediated mechanism for macromolecule uptake into the periplasm.”

‘- Also, it's worth noting that the statement that dextran binds to fibres that originate from the large crateriform structures does not exclude endocytosis, though I agree that their cryoEM argues against endocytosis.’

We agree with the Reviewer that our MC-mutant is no bulletproof disproof of endocytosis-like uptake. However, we added a cluster analysis (Fig. S11) of dSTORM experiments and demonstrated that there are no vesicles. We further provide CET data that was highly appreciated by Reviewer #4 and that again did not reveal a single vesicle. We undertook all these experiments to prove the existence of endocytosis but there is just no way to keep this hypothesis alive. However, we modified our manuscript to clarify that our mutant is no definitive proof as mentioned above:

Results:

“Thus, our results demonstrate that *P. limnophila* does not employ the MC protein Plim_1972 for dextran internalization.”

Discussion:

“However, since *P. limnophila* comprises more than one MC-like gene, we cannot exclude that our mutant does not affect the proposed function in vesicle formation. Taken together, we conclude that Planctomycetes do not use a vesicle mediated mechanism for macromolecule uptake into the periplasm.”

‘- In Fig. 1, I think I may be missing what they want me to notice. I understand the argument about the enlarged periplasmic space, but what is it they want me to notice in the phase contrast images (and why do they appear blue in my PDF? It makes it seem as if it's a DAPI overlay, but that can't be true). How am I supposed to see that the structures are connected to the OM?’

We agree with Reviewer #3 and modified the manuscript accordingly. The Phaco image was taken with a Nikon color camera and due to aberrations of the white LED at the phase ring, light appears bluish. However, since in *P. limnophila* internal structures are very difficult to see we removed this sentence from the manuscript.

‘- Similarly, I just don't see from the SR-SIM how I'm supposed to see that the membrane structures are periplasmic spaces - looks like GFP is actually inside these spaces in some cases (e.g., in B)’

We entirely revised Figure 2 (Figure 1 before revision) and provide more detailed information in the manuscript as well. Basically, red dots in WF can be better resolved in SR-SIM where invaginations are visible (Fig. 2 II e+f Nile Red). If compared to CET images, this becomes more obvious. We agree that with GFP as cytosolic marker the effect is not as strong as if the periplasmic space is labeled with dextran (e.g.

movies).

‘ - For the proteome measurements, seems odd that there are so few OM proteins (<1%) - what is the total abundance of OM proteins out of their spectral counts? I'm worried that they're missing a lot of material and hence the numbers are not so meaningful.’

The reviewer is completely right that at a first glance the percentage of outer membrane proteins identified in our approach seems to be very small. However, using the PSORT algorithm only 54 outer membrane proteins can be predicted to be encoded by the genome sequence of *Planctopirus limnophila* DSM 3776. Accordingly, by the identification of 23 of these proteins we covered 43% of the predicted subproteome. The average intensity/protein measured by mass spectrometry reached 1.02×10^8 for the outer membrane proteins and was thus 1.45 fold higher than the average intensity/protein for all proteins. Moreover, we must presume that the expression of outer membrane proteins is tightly regulated and therefore not all the predicted proteins might be amenable to analysis using one specific sample. Considering the fact that we took samples only at one time point during growth the total number of identified outer membrane proteins is thus within a realistic range.

‘ - I'm confused about the appendages (I don't think "appendices" is the word that they're meaning to use): I would think that if they are membrane-bound, then they would appear in light microscopy images. This brings up a larger point, that the recent history of papers about appendages (e.g. from Sigal Ben-Yehuda's lab on "nanotubes") have not been well received, with many suggesting that they are TEM artifacts. Would be good to have separate confirmation from light microscopy, which seems like it would be obvious based on their size.’

We agree with the Reviewer that bacterial appendages can result from artefacts during critical point drying of SEM specimens or through fixation in TEM analysis. Many –maybe even nanotubes- might be caused by such effects. However, we observe such structures in CET, a method that is artefact-free as our specimens are in a frozen hydrated stage. Please see Fig. 3 white arrowheads: empty crateriform structures. In panel (d) e.g. the most bottom right crateriform structure gives rise to a fiber. As we still do not know what they are we would prefer to keep the term appendages.

However, the argument with the light microscope is very good. The situation is a bit tricky: we could observe the stalk and holdfast structure of *P. limnophilus*, but not the fibers that originate from the large crateriform structures. However, we did observe the fibers from the large crateriform structures in *Gemmata*. Thus, we modified Figure 6 accordingly to provide additional light microscopic proof that our appendages exist and they are not artefacts from sample preparation.

‘ - As far as I can tell, the authors do not give info about the growth rates, nor which OD the cells are being imaged at in the Methods. This would be very useful knowledge.’

OD measurement is a tricky thing for Planctomycetes. They produce biofilms at the walls of the flasks and only the daughter swimmer cells live planktonic. Sometimes aggregates are formed between stalked mother

cells that interfere with OD measurements. Thus, we provided carefully chosen details regarding the cultivation conditions to ensure our experiments can be reproduced. This worked out in the past very well as e.g. our genetic system was reproduced several times in independent labs. We acknowledge the problem in a recent study (Jeske *et al.* *Frontiers in Microbiology* 2016).

‘Please note that I’m not suggesting that they need to go and knockout the other 9 genes to address the MC protein question. I think in the sum total of all of their experiments, they can certainly argue convincingly against endocytosis. However, it would be nice to have a richer analysis of these genes since I don’t think this is going to convince people who work on MC proteins that there is no endocytosis in Planctomycetes.’

As mentioned above the same scientist -Damien Devos- who discovered MCs in bacteria performed this analysis to ensure consistency. However, we agree with Reviewer #3 that this is no final proof and modified the manuscript as follows (new parts in yellow):

“This mutant lacks the homologue of the membrane coat-like protein that was proposed to localise at endocytic vesicles in *G. obscuriglobus*²⁰. However, since *P. limnophila* comprises more than one MC-like gene, we cannot exclude that our mutant does not affect the proposed function in vesicle formation. Taken together, we conclude that Planctomycetes do not use a vesicle mediated mechanism for macromolecule uptake into the periplasm.”

‘Some minor comments and questions:

- have the authors tried correlated SEM/SIM? This seems like it would help to address question above about the appendages’

Great suggestion, but we could only indirectly correlate results (Fig. 2 and Fig. S2 III) as we unfortunately lack access to a microscope which allows such analyzes with the same specimen. However, we would be very interested to apply this technique for future experiments and suggestions where to do so are very welcome!

‘- I’d like to see some discussion of the nucleoid compaction - are there similarities to other, more well studied organisms?’

For the sake of focus we did not address this topic in detail. However, we are currently preparing more CET analysis of *P. limnophila* and *Deinococcus radiodurans*. While very interesting, this is another story to be told later.

‘- In the introduction, there is a citation to "Rast et al, in prep" regarding a statement about the Planctomycetes having a proteinaceous cell wall; I assume the reference is a paper that shows that in fact the Planctomycetes have a PG cell wall. If so, I don’t think this statement belongs here, there is no evidence in the paper to support this conclusion and the previous paper has not been even submitted. If not, this citation doesn’t belong where it is, where it is modifying bacteria having PG cell walls.’

We have shown that planctomycetes possess a PG cell wall (Jeske et al. Nature Communications 2015). However, among free-living (not host associated) bacteria -besides Planctomycetes- only Verrucomicrobia of the subdivision 4 are thought to lack PG. The manuscript from Rast *et al.* shows that subgroup 4 Verrucomicrobia possess a PG cell wall as well. Thus, the citation was meant to proof that all free-living bacteria do possess a PG cell wall. However, the manuscript is in review and multiple additional experiments are asked for. Thus, it will most likely be published after this study. Consequently, we removed the citation and modified the statement as follows:

“For example, Planctomycetes were thought to have a proteinaceous cell wall^{12,13}, while a peptidoglycan cell wall to maintain cell integrity is the bacterial hallmark trait.”

‘- Would be useful to have a clearer description of the craters and what the authors are concluding: are they saying there are two types, and do these types differ as to whether the membrane is attached (and how do they determine membrane attachment?)’

As reviewer #4 pointed out: this will be a follow-up story. However, to clarify we modified Figure 6 to make clear that two different sorts of crateriform structures exist and that these structures sit in the outermost membrane. This is supported by CET data as well.

‘Reviewer #4 (Remarks to the Author):

Dear Authors of the manuscript NCOMMS-16-13455,

by a Senior Editor of Nature Communications I was asked to comment on the cryo-electron microscopy/tomography analyses included in your manuscript. Other parts of your research concerning planctomycetes biology would be reviewed by other referees; because the manuscript is under review for already some time I was asked to do it within 7 days (unfortunately, the Email is confidential - I am not allowed to cite it here as I originally did).

I agreed and got the manuscript 09/08/2016 10:42.

To keep the deadline (almost), I concentrated on the cryoEM part and can say: it fulfills all criteria of the world-renown top-level "Good (Baumeister) cryoEM-Laboratory Practice". More literature should be cited in "Materials and Methods" (as indicated in the annotated merged manuscript-file 98984_0_merged_pdf_1790422_m8gmzn_CH.pdf, that I send separately to the Editor - I thought that makes your life easier, and can/should be included in the new "transparent peer review system for new manuscripts submitted from January 2016" that I very much welcome/appreciate).’

We highly appreciate Reviewer #4's judgment and revised the manuscript accordingly, by adding the following citations:

- (1) Tivoli, W.F., Briegel, A. & Jensen, G. An improved cryogen for plunge freezing. *Microscopy and Microanalysis* 14, 375-379 (2008).
- (2) Schaffer, M. et al. Optimized cryo-focused ion beam sample preparation aimed at in situ structural studies of membrane proteins. *Journal of structural biology* (2016) <http://dx.doi.org/10.1016/j.jsb.2016.07.010>
- (3) Kremer J.B., Mastronarde, D.N. & McIntosh, J.R. Computer visualization of three-dimensional image data using IMOD. *Journal of structural biology* 116, 71-76(1996).

'I am now waiting for the next high-class paper from you showing subtomogram averaging data (from cryoET reconstructions of cryoFIB-SEM lamellae) of the "enigmatic" crateriform structures acting as buttons to connect inner and outer membrane (~100 per cell - it should be possible).'

Indeed, we are currently working on a detailed analysis of the crateriform structures, but it is still too early to talk about results.

'Here my comments along the Editor's criteria for reviewing:

1) What are the major claims of the paper?

They are all mentioned in the Abstract/Conclusions of the manuscript. (I annotated some proposals in the .pdf to strengthen it even further.)

2) Are they novel and will they be of interest to others in the community and the wider field?

They are revolutionary as they bring back Planctomycetes to their gram-negative bacterial roots (once again and even stronger). The study is not only important for microbiologists but also for everybody thinking/working in structural (cell) biology: it again shows that in particular emerging imaging technologies like cryoEM (here of thinned lamellae produced by cryoFIB-SEM) or superresolution (here 3DSIM) keeping the biological material in its best preservation state (vitreous for cryoEM) can help to correct wrong concepts. And that in an integrative manner building on newly developed genetic tools, additional (conventional) imaging techniques, proteomics and many more.

3) Is the work convincing, and if not, what further evidence would be required to strengthen the conclusions?

The paper already provides a wealth of convincing evidence (and appropriate statistics), nothing more is needed for the conclusions drawn.

4) On a more subjective note, do you feel that the paper will influence thinking in the field?

See point 2.

I like your manuscript very much, indeed. My recommendation to the Editor is: (very) minor revision and acceptance/publication asap.

*I do not send any "Confidential remarks for the editor only".
To the Editor: Thanks for giving me the opportunity.
Kind regards*

Christoph Hagen

(Oxford, 17/08/2016, ~1am)'

We highly appreciate Reviewer #4's assessment. Furthermore, we thank Reviewer #4 for his suggestions provided in the separate pdf file. If not otherwise mentioned here, we followed his suggestions:

1. We agree to move Figure S1 into the main manuscript. However, the combination with Figure 1 did not convince us. Thus, we move Figure S1 as separate new Figure 1 into the main manuscript while the entirely reworked Figure 1 is now Figure 2.
2. The DAPI-stained area in morphotypes II and III are not smaller in size. Due to the invaginations, the nucleoid has a different 3D structure. While details are beyond the scope of the current manuscript, we will further address the planctomycetal nucleoid architecture in a separate future study as described above.
3. The former Fig. 5 was entirely revised.

REVIEWERS' COMMENTS:

Reviewer #1 (Remarks to the Author):

My main concerns were mainly about the lack of quantification in the manuscript. These have largely been addressed and I appreciate the authors have made a decent attempt to quantify as much as they can. I would recommend the manuscript for publication.

Reviewer #3 (Remarks to the Author):

I apologize to the authors and editor for the delay in my response. I've reviewed the response from the authors, and they have addressed many of the major concerns that I had in the first submission. In particular, the inclusion of light microscopy images in Fig. 6 addresses the concern that the appendages are an artifact.

I think that the manuscript is worthy of publication in Nature Communications, though I do think the authors should address the following:

The authors responded to my previous comment about the abundance of OM proteins to say "The reviewer is completely right that at a first glance the percentage of outer membrane proteins identified in our approach seems to be very small. However, using the PSORT algorithm only 54 outer membrane proteins can be predicted to be encoded by the genome sequence of *Planctopirus limnophila* DSM 3776. Accordingly, by the identification of 23 of these proteins we covered 43% of the predicted subproteome. The average intensity/protein measured by mass spectrometry reached 1.02×10^8 for the outer membrane proteins and was thus 1.45 fold higher than the average intensity/protein for all proteins. Moreover, we must presume that the expression of outer membrane proteins is tightly regulated and therefore not all the predicted proteins might be amenable to analysis using one specific sample. Considering the fact that we took samples only at one time point during growth the total number of identified outer membrane proteins is thus within a realistic range."

This seems a strange argument: unless the authors can somehow argue that it makes sense that only 54 OM proteins would be identified for *P. limnophila*, then this seems to point more towards the entire comparison based on PSORT itself being faulty. Thus, I don't see how this falls in a realistic range, as the author's argue. It would strengthen the author's argument to either modify this part of the paper or address my concern above – I leave it to the editor to assess whether this is necessary for publication.

(Reviewer #3):

'I think that the manuscript is worthy of publication in Nature Communications, though I do think the authors should address the following:

*The authors responded to my previous comment about the abundance of OM proteins to say "The reviewer is completely right that at a first glance the percentage of outer membrane proteins identified in our approach seems to be very small. However, using the PSORT algorithm only 54 outer membrane proteins can be predicted to be encoded by the genome sequence of *Planctopirus limnophila* DSM 3776. Accordingly, by the identification of 23 of these proteins we covered 43% of the predicted subproteome. The average intensity/protein measured by mass spectrometry reached 1.02×10^8 for the outer membrane proteins and was thus 1.45 fold higher than the average intensity/protein for all proteins. Moreover, we must presume that the expression of outer membrane proteins is tightly regulated and therefore not all the predicted proteins might be amenable to analysis using one specific sample. Considering the fact that we took samples only at one time point during growth the total number of identified outer membrane proteins is thus within a realistic range."*

*This seems a strange argument: unless the authors can somehow argue that it makes sense that only 54 OM proteins would be identified for *P. limnophila*, then this seems to point more towards the entire comparison based on PSORT itself being faulty. Thus, I don't see how this falls in a realistic range, as the author's argue. It would strengthen the author's argument to either modify this part of the paper or address my concern above – I leave it to the editor to assess whether this is necessary for publication.'*

We agree with Reviewer #3 that this section of our manuscript was still misleading. We reanalysed the dataset and revised the entire paragraph accordingly. For detailed clarification:

First we predicted outer membrane proteins (OM) employing the PSORT algorithm based on the genome sequence of *P. limnophila*. We obtained 31 putative outer membrane proteins, not the 54 mentioned before. The number 54 was the result of a duplication in the NCBI proteome database for *P. limnophila* as the organism was recently renamed.

This number was then compared to PSORTdb (new additional Ref. Paebody et al. 2015) a database that includes genomes of 5042 Gram-negatives that possess an outer membrane. For these genomes in total 175500 OM were predicted by PSORT (Paebody et al. 2015). Thus, an average Gram-negative bacterium comprises 34,8 OMs ($175500 / 5042$) according to PSORTdb (Paebody et al. 2015). Since we detected 24 OMs employing our LC-MS approach we covered 77%, which is in the range our proteome experts (Martin Kucklick and Susanne Engelmann) would expect for typical Gram-negative bacteria. Reviewer #3 might be right that the number falls short compared to other prediction methods. However, in this study we only compare PSORT results and thus the comparison between Planctomycetes and other Gram-negatives is valid from our perspective.

Furthermore, in the revised version of our manuscript we make the point that we found other envelope proteins such as periplasmic proteins as well. Until now it is believed that Planctomycetes entirely lack an outer membrane. Thus, the detection of 24 outer membrane proteins and 34 periplasmic proteins in the proteome membrane fraction provides in our opinion a strong argument for the Gram-negative cell architecture of Planctomycetes.

However, we agree with Reviewer #3 that *P. limnophila* and other Planctomycetes might differ in their outer membrane proteome composition from other Gram-negatives. This is mainly due to the unique crateriform structures that are exclusively found among Planctomycetes and thus might not be detected with PSORT. Details will be included in a follow up study on crateriform structures.